# Sea ice dynamics across the Mid-Pleistocene transition in the Bering Sea

H. Detlef [1], S.T. Belt [2], S.M. Sosdian [1], L. Smik[2], C.H. Lear[1], I.R. Hall [1], P. Cabedo-Sanz[2], K. Husum [3] & S. Kender[4,5]

Sea ice and associated feedback mechanisms play an important role for both long- and short-term climate change. Our ability to predict future sea ice extent, however, hinges on a greater understanding of past sea ice dynamics. Here we investigate sea ice changes in the eastern Bering Sea prior to, across, and after the Mid-Pleistocene transition (MPT). The sea ice record, based on the Arctic sea ice biomarker $IP_{25}$ and related open water proxies from the International Ocean Discovery Program Site U1343, shows a substantial increase in sea ice extent across the MPT. The occurrence of late-glacial/deglacial sea ice maxima are consistent with sea ice/land ice hysteresis and land−glacier retreat via the temperature−precipitation feedback. We also identify interactions of sea ice with phytoplankton growth and ocean circulation patterns, which have important implications for glacial North Pacific Intermediate Water formation and potentially North Pacific abyssal carbon storage.

[1] School of Earth and Ocean Sciences, Cardiff University, Cardiff CF10 3AT, UK. [2] School of Geography, Earth and Environmental Sciences, Plymouth University, Plymouth PL4 8AA, UK. [3] Norwegian Polar Institute, Fram Centre, Tromsø 9296, Norway. [4] Camborne School of Mines, University of Exeter, Penryn, Cornwall TR10 9EZ, UK. [5] British Geological Survey, Keyworth, Nottingham NG12 5GD, UK. Correspondence and requests for materials should be addressed to H.D. (email: DetlefH1@cardiff.ac.uk)

Sea ice plays a key role in both long-term[1,2] and abrupt millennial-scale[3] climate change as a result of its far-reaching climate feedbacks, including the ice albedo effect, ocean-atmosphere gas/moisture exchange and ocean circulation patterns. However, it is only through the identification of long-term sea ice dynamics that our understanding of the role of sea ice for climate change can improve and hence our ability to predict future sea ice extent.

In modern times, sea ice in the Bering Sea forms in the Chukchi Sea and in polynyas along the southward facing coastlines on the eastern Bering Sea shelf. Thereafter, sea ice is advected south-westward[4], reaching its maximum extent approximately at the shelf edge (Fig. 1). Nutrient-release during the spring sea ice melting, eddy-driven upwelling and shelf edge processes sustain a diverse ecosystem with high primary productivity, especially along the eastern Bering Sea slope, often referred to as the 'Green Belt'[5,6,7]. In general, the subarctic North Pacific sea ice regime plays an important role in North Pacific Intermediate Water (NPIW) formation as a result of brine rejection during sea ice freezing. Today, NPIW is formed in the Sea of Okhotsk, whereas glacial NPIW (GNPIW) was at least partly formed in the Bering Sea[8–11]. Increased ventilation and extent of GNPIW during cold phases[11,12] demonstrates the importance of the Bering Sea for glacial oceanic circulation patterns in the North Pacific realm and beyond[12]. Investigating the interactions of sea ice dynamics with ocean circulation and productivity patterns and identifying the role of sea ice for major climate transitions is critical for our understanding of Arctic and sub-Arctic climate. One such climate transition during the Quaternary Period is the Mid-Pleistocene transition (MPT, 1.2 −0.7 Ma).

The MPT marks a fundamental shift in frequency and amplitude of northern hemisphere glaciations from 41 ka glacial/interglacial (G/IG) cycles to quasi-periodic glaciations at 100 ka[13–16], yet the change in G/IG frequency occurs without any attributable change in orbital forcing. Thus, the MPT marks a shift in the response of the climate system to orbital forcing, likely caused by internal climate mechanisms. Conceptual modelling

has identified potential key feedback mechanisms involving sea ice, such as the so-called 'sea ice-switch' hypothesis (SIS)[1,2,17], which suggests that sea ice can modify the climate state, switching it between a growing and a retreating land glacier mode, via a temperature–precipitation feedback. This hypothesis makes two critical predictions. First, the SIS invokes a gradual deep ocean cooling and change in ocean vertical mixing as the underlying cause for increased high-latitude sea ice extent during the MPT[1,2,17]. Second, the SIS proposes a land versus sea ice hysteresis, with large sea ice extent across early deglaciations[2]. Recent modelling studies also suggest that the periodicity of G/IG cycles is linked to changes in the interhemispheric pattern of sea ice growth[18]. However, while modelling studies clearly suggest the likely importance of sea ice for controlling climate change across the MPT[1,2,17,18], complementary high-resolution proxy-based reconstructions of sea ice dynamics are yet to be reported.

Recent advances in the development of source-specific biomarkers for paleoenvironmental reconstructions, including $IP_{25}$ (Ice Proxy with 25 carbon atoms)[19], a proxy for seasonal Arctic sea ice, together with those indicative of open water conditions[20] and in combination with proxies for primary productivity[21], such as the mass accumulation rate of biogenic opal ($MAR_{opal}$), enable high-resolution reconstructions of past sea ice dynamics. The $IP_{25}$ biomarker is a mono-unsaturated highly branched isoprenoid (HBI) lipid produced by certain sea ice diatoms during spring[22,23], providing proxy evidence for past seasonal sea ice. To date, $IP_{25}$ has been readily identified within sediments dating back to the Plio-Pleistocene boundary[24] and has even been detected in sediments of Miocene age[25], albeit from different locations. Sedimentary $IP_{25}$ abundance has been found to reliably reflect variations in seasonal sea ice extent, while absent/low $IP_{25}$ is normally considered to reflect ice-free or extended sea ice cover regimes[26–32]. Consistent with these interpretations, $IP_{25}$ is present in surface sediments from sites in the sub-polar North Pacific that experience seasonal sea ice cover during modern times, but is absent from year-round ice-free locations[32]. Further, elevated concentrations of a tri-unsaturated HBI biomarker (HBI III), shown recently to be produced by certain diatoms in polar

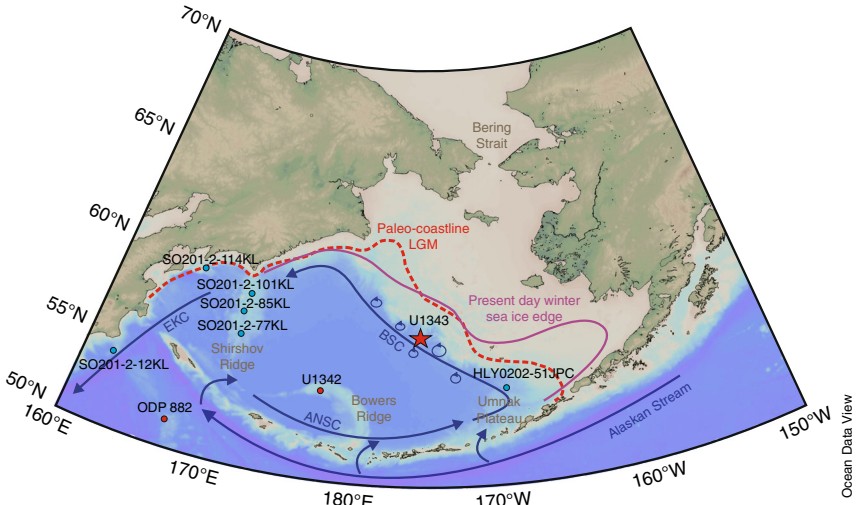

**Fig. 1** Overview map of the study location. Map of the Bering Sea showing the location of IODP Site U1343 (red star), U1342 and ODP Site 882 (red dots), together with additional sediment cores (light blue dots) where sea ice reconstructions were performed across Termination I (SO201-2-12KL, SO201-2-77KL, SO201-2-85KL, SO201-2-101KL, SO201-2-114KL[21, 67], HLY0202-51JPC[44]). The pink solid line represents the present day winter sea ice edge, the red dashed line is the Last Glacial Maximum (LGM) coastline[68], and the blue arrows show the cyclonic surface circulation in the Bering Sea. The Alaskan Stream enters the Bering Sea in the south through several Aleutian passes to form the Aleutian North Slope Current (ANSC) that feeds into the Bering Slope Current (BSC), which in turn feeds into the East Kamchatka Current (EKC) (modified from Stabeno et al.[60]). Eddy formation along the eastern Bering slope (blue circles) brings nutrients to the surface resulting in high primary productivity, called the 'Green Belt'[7]. Map created with Ocean Data View[69]

environments[33], reflect the spring ice-edge bloom within the open waters of the marginal ice zone (MIZ), at least within the Barents Sea[20], which has a similar annual sea ice cycle to the Bering Sea. In contrast, lower abundances of HBI III[20] and variable abundances of other phytoplankton biomarkers, including brassicasterol[34], are found in year-round ice-free settings of the Barents Sea and Norwegian Sea. However, lower abundances of phytoplankton biomarkers also generally occur under perennial sea ice conditions, similar to IP$_{25}$. Measurement of MAR$_{opal}$ provides an alternative means of distinguishing these two extremes in sea ice cover, especially as siliceous phytoplankton are the most important primary producers in the Bering Sea today[35]. In the subarctic North Pacific, MAR$_{opal}$ represents first-order changes in primary productivity[36], such that extended sea ice cover leads to decreased productivity in the region[36]. Previously, Méheust et al.[21] used the sedimentary biogenic opal content in the western Bering Sea across the Last Glacial Maximum to Holocene transition (Termination I), in order to distinguish between different sea ice states.

Additionally, beyond looking at the absolute biomarker concentrations, the nature of the correlation between IP$_{25}$ and HBI III can also provide insight into seasonal sea ice dynamics, although these relationships are not yet fully understood. Thus, observational surface sediment calibration studies[20,37] and downcore records[20,38] have shown that a weak or inverse relationship between IP$_{25}$ and HBI III is associated with a strong seasonal sea ice cycle, whereas a positive in-phase relationship likely reflects a fluctuating sea ice margin, with reduced seasonality, and smaller changes in the position of the winter and summer sea ice edge. Other biomarkers, including brassicasterol, are also indicative of open water settings[29], although the complication of other potential sources (e.g. riverine input and potentially sea ice algae[39]), somewhat limits their use beyond a qualitative indication of general phytoplankton production. However, the relationship between IP$_{25}$ and brassicasterol can provide context with respect to phytoplankton production in the high-productivity region of the eastern Bering Sea slope[6].

At present, the only reconstruction of sea ice variability from the subarctic North Pacific spanning the MPT is based on sea ice diatoms from International Ocean Discovery Program (IODP) Site U1343[40] (Supplementary Fig. 1), which is located off the eastern Bering slope, close to the present day winter sea ice margin (57°33.4′N, 176°49.0′W; 1950 m, Fig. 1)[41]. While this study suggests an overall increase in sea ice extent across the MPT, its low temporal resolution (~15 ka), precludes a robust evaluation of sea ice dynamics on G/IG timescales. Here, we address this gap by presenting a high-resolution sea ice reconstruction for the MPT, between 1.22 and 0.8 Ma, from IODP Site U1343. We further examine our MPT findings by comparisons with data obtained from the same core (Site U1343) corresponding to the pre-MPT 41 ka (~1.53–1.36 Ma) and late Pleistocene 100 ka (0.50–0.34 Ma) G/IG cycles to test previous hypotheses of a strong causal link between sea ice and the changing nature of G/IG cycles. Our reconstruction is based on IP$_{25}$ from Site U1343, together with HBI III and MAR$_{opal}$[36]. Building on the approach of Méheust et al.[21], we utilise MAR$_{opal}$ with threshold concentrations of IP$_{25}$ and HBI III (see Methods) in order to provide a classification of the sea ice states recorded at Site U1343. We identify a substantial increase in sea ice in the Bering Sea and the appearance of transient late-glacial/deglacial sea ice maxima across the MPT, in support of land glacier retreat via a temperature–precipitation feedback mechanism. Together with existing regional and global climate records, we additionally propose that sea ice extent in the eastern Bering Sea plays an important role for GNPIW formation and potentially North Pacific carbon storage.

## Results

**Marine isotope stages 51 to 44.** Marine isotope stages (MIS) 51 to 44 represent the time interval from 1.53 to 1.36 Ma and thus precede the onset of the MPT at around 1.2 Ma. The benthic foraminifera oxygen isotope ($\delta^{18}O_b$) record from Site U1343 is of low temporal resolution between 2.4 and 1.2 Ma (~10 ka). Wavelet analysis of the U1343 $\delta^{18}O_b$[42], however, shows significant 41-ka periodicity from 1.5 Ma in accordance with the global $\delta^{18}O_b$ (LR04) stack[16]. Although the relative timing of sea ice changes in comparison to G/IG cycles is uncertain, resulting from the low resolution of the U1343 $\delta^{18}O_b$ record prior to 1.2 Ma (Fig. 2), it is nonetheless clear that this interval is within the late Pleistocene as indicated by biostratigraphy, magnetostratigraphy, and tuning of long-term U1343 $\delta^{18}O_b$[42] to the LR04 stack[16].

IP$_{25}$ was identified in 26 out of 32 samples analysed during MIS 51 to 44 (Fig. 2), demonstrating the presence of seasonal sea ice in the eastern Bering Sea prior to the MPT. Periods of absent IP$_{25}$ were accompanied by variable MAR$_{opal}$ values indicative of both ice-free (high MAR$_{opal}$) and perennial sea ice (low MAR$_{opal}$) conditions (Fig. 2, Supplementary Fig. 1). Concentrations of HBI III were generally low, indicating that a spring ice edge bloom associated with the retreating MIZ most likely did not occur in the vicinity of Site U1343 during this interval. Further, a weak positive relationship between IP$_{25}$ and HBI III (IP$_{25}$-HBI III: $r_{xy} = 0.47$ with 95% Student's $t$ confidence intervals [0.176; 0.688], $n = 32$) is indicative of a fluctuating sea ice margin, with smaller changes in the position of the summer and winter sea ice edge.

**Marine isotope stages 36 to 20.** Our most detailed interval spans MIS 36 to 20 (1.22–0.8 Ma), thus covering the onset and the majority of the MPT (1.2–0.7 Ma). Wavelet analysis of Site U1343 $\delta^{18}O_b$ indicates dominant 100-ka periodicity of G/IG cycles from 0.7 Ma[42].

IP$_{25}$ was identified in 71 out of the 78 samples analysed, demonstrating the presence of seasonal sea ice in the eastern Bering Sea throughout most of the MPT, consistent with the observations from MIS 51 to 44. IP$_{25}$ concentration exhibits distinct variability on G/IG timescales, with increased values throughout most glacial intervals (Fig. 2), indicative of regionally enhanced sea ice cover. Low IP$_{25}$, HBI III and MAR$_{opal}$ values during MIS 35, 26/25 and 22 collectively suggest extended sea ice cover at the eastern Bering slope during these intervals (Fig. 2, Supplementary Fig. 1). In contrast, low IP$_{25}$ and HBI III concentrations, together with a relatively high MAR$_{opal}$ during MIS 31 and 33, suggest ice-free conditions (Fig. 2, Supplementary Fig. 1). However, the biomarker values at ~0.85 Ma (MIS 21) and ~1.18 Ma (early MIS 35) (Table 1) do not convincingly fall within the proposed sea ice state classifications (Table 2), largely as a result of relatively high HBI III content (>0.7 ng g$^{-1}$ dry sediment (sed)). The reasons for this are unclear, but potentially indicate the influence of occasional MIZ sedimentation.

A pronounced increase in peak glacial IP$_{25}$ values is observed consistently across all glacial periods studied post MIS 28 (~1.0 Ma) concomitant with a shift in the timing of the glacial IP$_{25}$ maximum from the mid-glacial to the late-glacial/deglacial (Fig. 2), with exception of MIS 22. IP$_{25}$ maxima throughout the entire interval are accompanied by low MAR$_{opal}$ values in Site U1343 (Fig. 2), indicating reduced primary productivity as a result of seasonally increased sea ice cover. HBI III concentration is relatively low prior to MIS 31, before increasing thereafter (Fig. 2). In fact, the highest HBI III concentrations throughout all three intervals are observed from MIS 28 to MIS 26. Although the timing of the HBI III peaks throughout G/IG cycles is variable,

higher values are generally observed during glacial intervals (Fig. 2).

Notably, the correlation of IP$_{25}$ and HBI III changes to one that is in-phase between 1 and 0.95 Ma (Fig. 3) ($r_{xy}$ = 0.665 with 95% Student's $t$ confidence intervals [0.145; 0.897], $n$ = 17), indicating a more fluctuating sea ice margin with predominantly MIZ conditions. This is framed by intervals of no apparent correlation between these two biomarkers from 1.22 to 1.0 Ma, MIS 36 to 29 ($r_{xy}$ = −0.016 with 95% Student's $t$ confidence intervals [−0.296; 0.267], $n$ = 26) and 0.95−0.8 Ma, MIS 24 to 20 ($r_{xy}$ = −0.174 with 95% Student's $t$ confidence intervals [−0.438; 0.117], $n$ = 35).

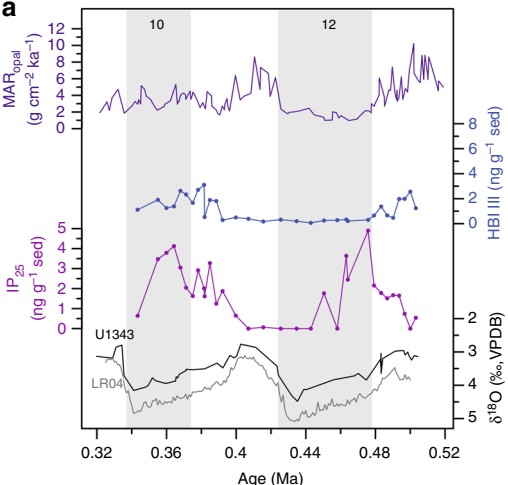

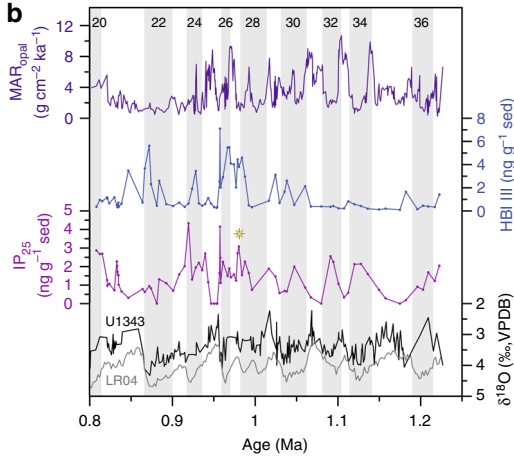

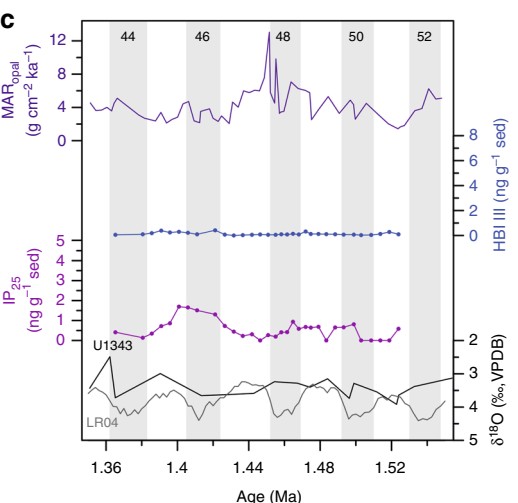

(Fig. 3) more consistent with a stronger seasonal cycle and a pronounced advance and retreat of the sea ice margin, as seen in the modern setting and for most other Arctic marginal seas[32,43].

**Marine isotope stages 13 to 10**. The youngest interval in this study (MIS 13 to 10; 0.5−0.34 Ma) represents two G/IG cycles during the post-MPT quasi-100 ka climate variability.

IP$_{25}$ was quantified in 26 out of 32 samples analysed for biomarkers. A broad gradual increase of IP$_{25}$ and HBI III, accompanied by a high MAR$_{opal}$, occurs during cooling phases of the interglacial intervals, indicative of seasonal sea ice of increasing duration, with predominantly MIZ conditions (Fig. 2, Supplementary Fig. 1). Biomarker data at 0.5 Ma (MIS 13) do not conform to the suggested sea ice state classification (Table 1, Table 2) as a result of relatively high HBI III concentrations (>0.7 ng g$^{-1}$ sed), yet low IP$_{25}$ content (Table 1). However, a high MAR$_{opal}$ is indicative of ice-free conditions (Table 1, Supplementary Fig. 1). Maximum IP$_{25}$ values are reached during early glacial MIS 12 and MIS 10 (Fig. 2). Absent or very low IP$_{25}$ across the MIS 12/11 transition is accompanied by low HBI III concentrations, and low/high MAR$_{opal}$ values, indicating extended sea ice cover across the late-glacial/deglacial, and ice-free conditions during the interglacial MIS 11 from ~425 ka (Fig. 2, Supplementary Fig. 1). No change is observed in HBI III concentration during late MIS 12 and across the MIS 12/11 transition potentially due to the rapid transition from perennial to ice-free conditions across the termination. The relationship between IP$_{25}$ and HBI III shows no correlation across G/IG cycles ($r_{xy}$ = 0.236 with 95% Student's $t$ confidence intervals [−0.525; 0.787], $n$ = 32) consistent with the late MPT interval (0.95−0.8 Ma).

## Discussion

Variations in sea ice (IP$_{25}$) and phytoplankton (HBI III) biomarkers, together with previously reported MAR$_{opal}$ from Site U1343, demonstrate major changes in sea ice dynamics in the eastern Bering Sea during the past 1.5 Ma, especially across the interval covering the MPT (Fig. 4). Three broad intervals of sea ice change are identified corresponding to the early-mid Pleistocene (1.22−1.0 Ma), an interim state (1.0−0.95 Ma), and the mid-to-late Pleistocene (0.95−0.8 Ma and 0.5−0.34 Ma).

During the early-mid Pleistocene (1.22−1.0 Ma) IP$_{25}$ and HBI III concentrations are de-coupled (Fig. 3), indicating a pronounced seasonal advance and retreat of the sea ice margin (Fig. 5), as per modern conditions. This contrasts the preceding interval (MIS 51 to 43) where IP$_{25}$ and HBI III concentrations exhibit a weak positive correlation indicative of a fluctuating sea ice margin[38] and relatively small changes in the position of the winter and summer sea ice edge during 41-ka climate cycles, potentially as a result of less extreme climate variability (Fig. 2).

**Fig. 2** Bering Sea biomarker patterns across the three studied Pleistocene intervals. Overview of the three time intervals analysed for biomarkers. **a** 0.34−0.5 Ma, **b** 0.8−1.22 Ma and **c** ~1.36−1.53 Ma. Sedimentary mass accumulation rate of biogenic opal (MAR$_{opal}$) in Site U1343 is in purple[36]. IP$_{25}$ (violet) and HBI III (blue) in Site U1343 are expressed in ng g$^{-1}$ dry sediment (sed). The benthic foraminiferal oxygen isotope record of Site U1343[42] is in black together with the LR04 stack[16] in grey. The light grey vertical bars indicate glacial intervals and the white bars characterise interglacials (numbers at the top correspond to MISs, MIS boundaries from Lisiecki and Raymo[16]). The yellow asterisk indicates the first glacial interval where a late-glacial/deglacial sea ice maximum is observed (MIS 28)

**Table 1 Data points that do not fit the sea ice state classification**

| Age (Ma) | IP$_{25}$ (ng g$^{-1}$ sed) | HBI III (ng g$^{-1}$ sed) | MAR$_{opal}$ (g cm$^{-2}$ ka$^{-1}$) |
|---|---|---|---|
| 0.50 | 0 | 2.56 | 7.74 |
| 0.85 | 0.31 | 3.48 | 1.26 |
| 1.18 | 0.19 | 1.66 | 3.17 |

**Table 2 Boundaries for identification of sea ice states**

| Sea ice state | IP$_{25}$ (ng g$^{-1}$ sed) | HBI III (ng g$^{-1}$ sed) | MAR$_{opal}$ (g cm$^{-2}$ ka$^{-1}$) |
|---|---|---|---|
| Ice free | <0.5 | <0.7 | >4 |
| Extended sea ice | <0.5 | <0.7 | <4 |
| Seasonal sea ice (within the MIZ) | >0.5 | >0.7 | Variable |
| Seasonal sea ice (outside the MIZ) | >0.5 | <0.7 | Variable |

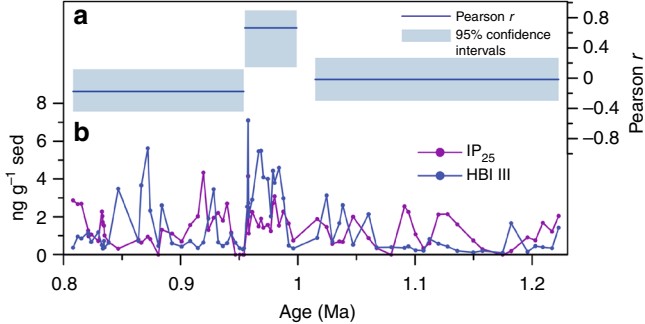

**Fig. 3** Correlation of IP$_{25}$ and HBI III across the MPT interval. **a** Pearson's $r$ correlation coefficient for IP$_{25}$ and HBI III between 0.8 and 1.22 Ma, including the 95% confidence interval (shaded blue area), calculated using Pearson T3[66]. The interval has been divided into three subsections based on the correlation of both biomarkers, the absolute concentration and the profile of sea ice increase across glacial periods. **b** IP$_{25}$ (violet) and HBI III (blue) concentrations from 0.8 to 1.22 Ma expressed in ng g$^{-1}$ sed

At ~1.0 Ma, an increase in IP$_{25}$ and HBI III, together with a shift to a statistically significant in-phase relationship between these two biomarkers (Fig. 3), indicates a return to a more fluctuating sea ice margin[38] and the onset of an interim state in sea ice dynamics. Highest HBI III concentrations within the entire record (Fig. 4) indicate that MIZ conditions likely prevailed close to Site U1343 (Fig. 5). Additionally, the temporal profile of sea ice increase across glacial periods changes during the interim state, such that sea ice reaches its maximum during the late-glacial/deglacial, compared to the mid-glacial IP$_{25}$ peaks of the early-MPT interval (1.22–1.0 Ma) (Fig. 4). The gradual early glacial increase in sea ice during the interim state is followed by a pronounced maximum in IP$_{25}$ concentration, first observed during MIS 28 (Fig. 4). Determination of the exact duration of late-glacial/deglacial sea ice maxima is limited by variable sample resolution, although the best resolved maximum during late MIS 26/early MIS 25 has a duration of ~4.8 ka. These transient late-glacial/deglacial sea ice maxima, even though of variable resolution (one to four data points), are observed during MIS 28, 26, 24 and 12. Compared to the sea ice maxima observed across the

MPT interval, however, the sea ice maximum during late MIS 12 is characterised by a prolonged period (~33 ka) of extended sea ice cover, as shown by the low MAR$_{opal}$ values (Fig. 4, Supplementary Fig. 1). This potentially points towards an intensification of late-glacial/deglacial sea ice maxima between the MPT G/IG cycles and the quasi-100 ka cycles of the late Pleistocene, though a continuous sea ice record, spanning at least the last 1.5 Ma, is needed, to confirm this suggestion. Nevertheless, in support of our interpretation, it has been reported previously that perennial sea ice dominated the western Bering Sea[21] and the Umnak Plateau[44] (Fig. 1) until ~15 ka BP across Termination I, with ice-free conditions not reached until ~11 ka BP (Fig. 5). Studies from other marginal seas, such as the Barents Sea, the Fram Strait and the Nordic Seas, also confirm the presence of an extensive sea ice cover during Termination I[3,30,45], suggesting that this could be a common feature of late Pleistocene climate cycles.

In addition to IP$_{25}$ and HBI III we measured a third biomarker, brassicasterol (24-methylcholesta-5,22E-dien-3β-ol), which has a variety of sources, including marine and lacustrine phytoplankton and potentially even sea ice algae[39]. As such, we use brassicasterol to investigate the importance of sea ice for phytoplankton growth in the eastern Bering Sea and, indeed, we find a weak positive relationship ($r_{xy} = 0.398$ with 95% Student's $t$ confidence intervals [0.151; 0.598], $n = 58$) between IP$_{25}$ and brassicasterol (Supplementary Fig. 2) from 1.53 to 1.36 Ma and from 1.2 to 1.0 Ma. This likely reflects production of brassicasterol by sea ice algae or phytoplankton stimulated by nutrient release during spring sea ice melting[46]. From ~1.0 Ma onwards, however, brassicasterol and IP$_{25}$ are de-coupled ($r_{xy} = -0.005$ with 95% Student's $t$ confidence intervals [−0.288; 0.278], $n = 84$) (Supplementary Fig. 2), which is suggestive of additional sources for brassicasterol (e.g. possibly from non-biogenic entrainment in sea ice) and/or other mechanisms potentially becoming more important for nutrient supply to the surface ocean at the eastern Bering Sea slope. Possible mechanisms include changes in the ocean stratification/vertical mixing and/or changes in the inflow of nutrient-rich Pacific waters, specifically the Alaskan Stream[47]. The abundance of the marine diatom species *Neodenticula seminae*[40], a proxy of Alaskan Stream inflow into the Bering Sea, shows a pronounced decrease in the mean percentage in Site U1343 at the end of the interim state ~0.95 Ma (Fig. 4). This suggests that mid-to-late Pleistocene glacial periods (at least) are characterised by decreased influence of North Pacific waters at the core Site as a result of glacial sea level lowstands[40]. With respect to nutrient transport to the surface ocean along the Bering Sea slope, this indicates that upwelling of nutrient-rich waters via deep-reaching eddies[7] potentially played the most important role for nutrient supply and phytoplankton growth from at least ~0.95 Ma onwards.

The continued inflow of North Pacific waters (as indicated by *N. seminae* (Fig. 4)) into the southern Bering Sea during the interim state (1.0–0.95 Ma) is consistent with the in-phase relationship between IP$_{25}$ and HBI III (Fig. 3), suggesting smaller variations in the position of the summer and winter sea ice margin. Even though regional climate cooling in the North Pacific from ~1.15 Ma (see below) promoted an increase in the eastern Bering Sea seasonal sea ice cover, as seen by increased IP$_{25}$ concentrations, the continued inflow of North Pacific waters could have counteracted an extensive seasonal expansion of sea ice and resulted in the observed fluctuating sea ice margin at Site U1343 across the interim state. Concomitant with the decrease in Alaskan Stream inflow at ~0.95 Ma, as suggested by the abundance of *N. seminae* in U1343 (Fig. 4), the correlation of HBI III and IP$_{25}$ shifts (Fig. 3) to one that is more indicative of a pronounced seasonal cycle in the position of the winter and summer sea ice margin.

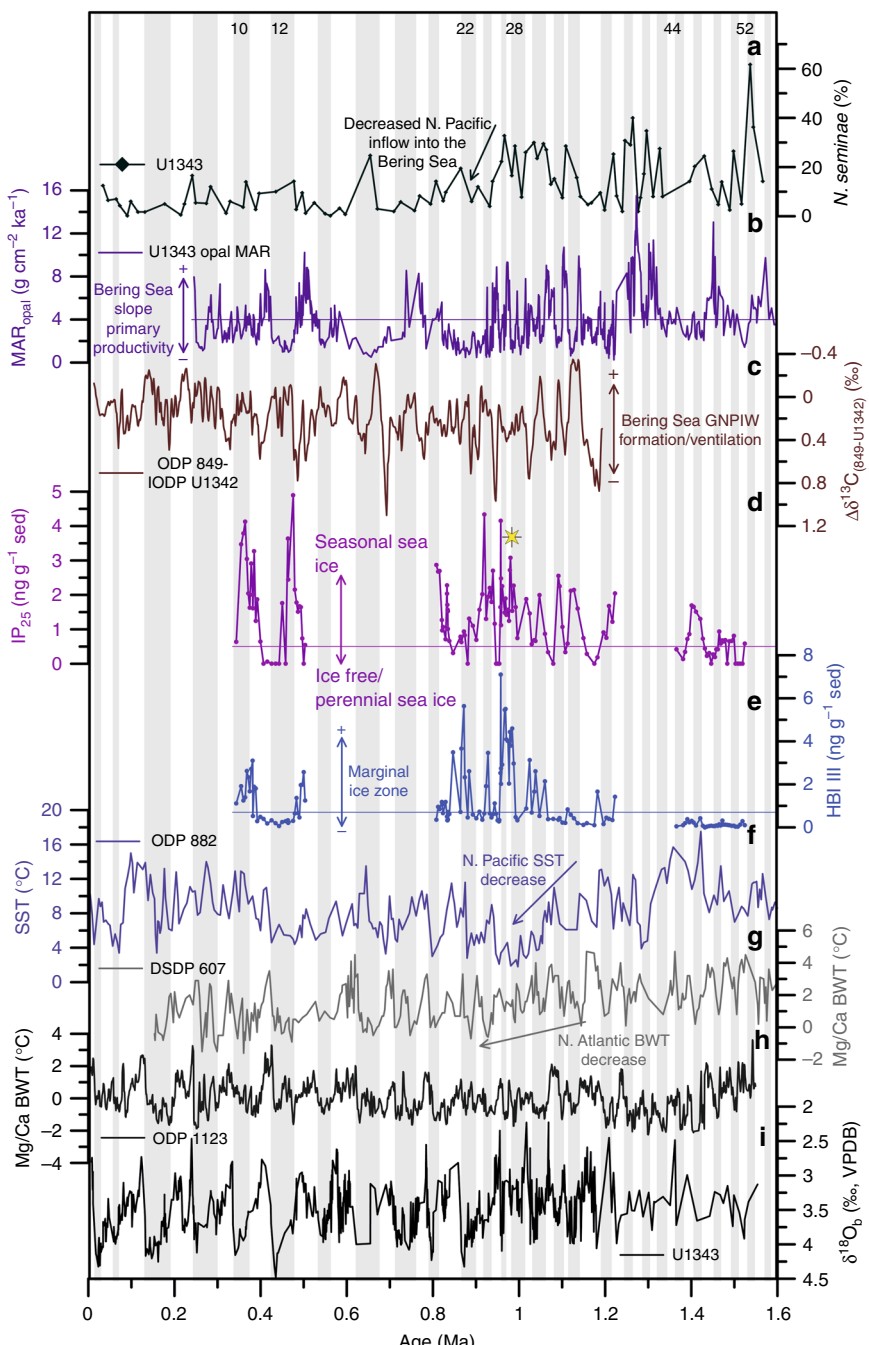

**Fig. 4** Sea ice dynamics in the Bering Sea in comparison with regional and global climate records. Regional and global climate records across the Pleistocene. **a** Abundance of *N. seminae* in sediments of Site U1343[40] (dark green). **b** Site U1343 sedimentary MAR$_{opal}$[36] (purple). **c** $\Delta\delta^{13}C_{(849\text{-}U1342)}$[10] indicative of GNPIW ventilation/formation (brown). **d** Site U1343 IP$_{25}$ (violet) record (this study). The IP$_{25}$ peak labelled with an asterisk denotes the first glacial interval (MIS 28) characterised by a late-glacial/deglacial sea ice maximum. **e** Site U1343 HBI III (blue) record (this study). **f** Alkenone-based North Pacific SSTs from ODP Site 882[51] (light purple). **g** BWT from DSDP Site 607[48] in the North Atlantic (light grey). **h** BWT from ODP Site 1123[50] in the southwest Pacific (dark grey). **i** Site U1343 $\delta^{18}O_b$ record (black)[42]. Grey bars indicate glacial intervals, white bars represent interglacials (numbers at the top correspond to MIS, MIS boundaries from Lisiecki and Raymo[16]). The horizontal lines represent the boundaries for MAR$_{opal}$, IP$_{25}$ and HBI III used to define sea ice states

The onset of the mid-to-late Pleistocene interval (0.95−0.8 Ma, 0.5−0.34 Ma), beginning at MIS 25, is characterised by an ice-free eastern Bering Sea (Fig. 4), as shown by absent IP$_{25}$, low HBI III and high MAR$_{opal}$. Temporal IP$_{25}$ variability across G/IG cycles is similar to that of the interim state with higher IP$_{25}$ concentration indicative of increased sea ice cover during the late-glacial/ deglacial. Exceptionally, MIS 22 is characterised by lower IP$_{25}$ values compared to the three preceding glacial periods (Fig. 4).

However, a consistently low MAR$_{opal}$ between MIS 23 and late MIS 21 suggest a persistent seasonal/extended sea ice cover during this time interval, which may explain the relatively lower IP$_{25}$ content during MIS 22.

Overall, our data suggest a twofold change in sea ice dynamics across the MPT with an increase in sea ice extent from ~1.15 Ma accompanied by a change in the timing of glacial sea ice increase at ~1.0 Ma. From 1.22 to 1.0 Ma sea ice maxima are encountered

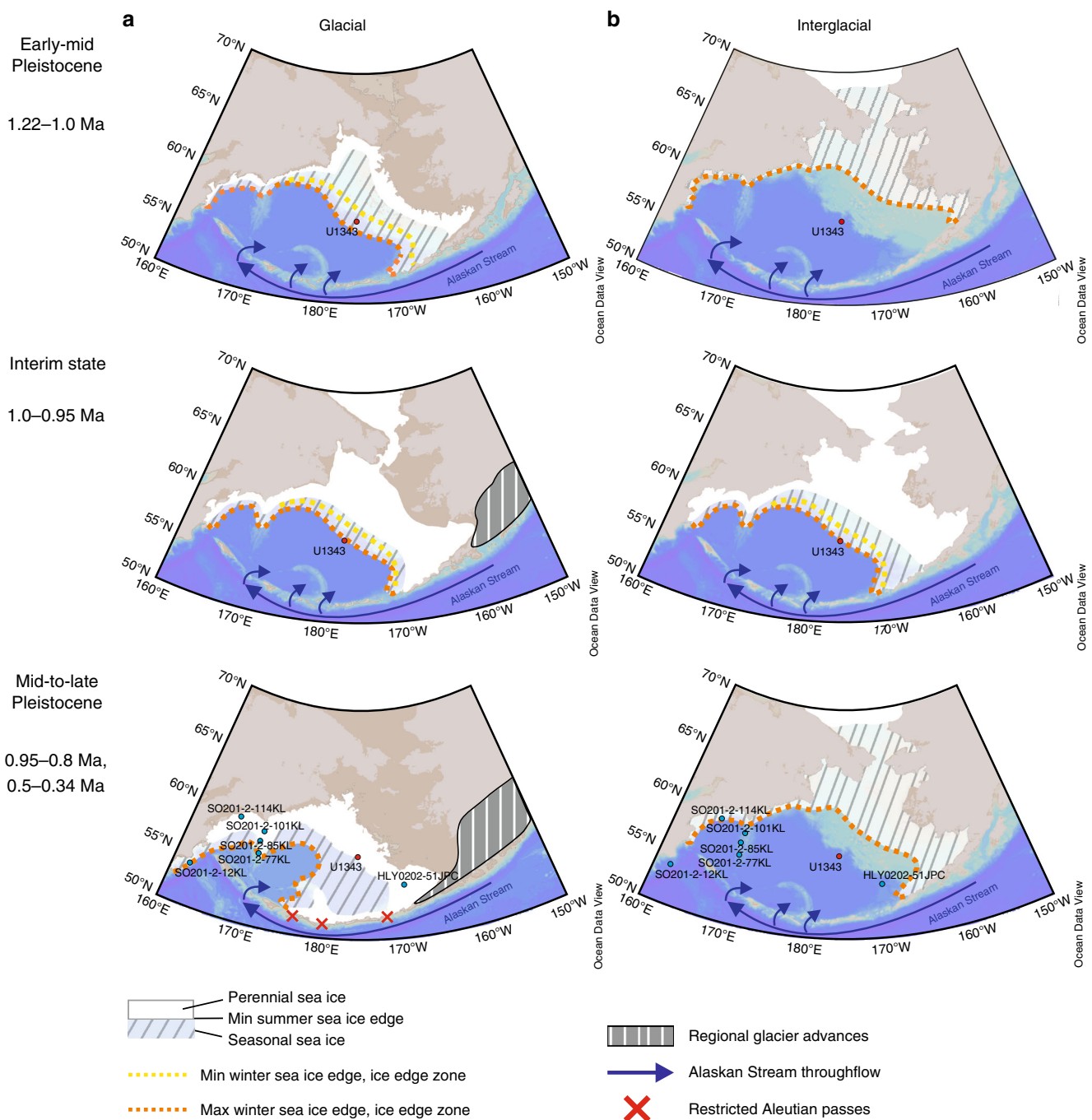

**Fig. 5** Schematic overview of the twofold change in sea ice dynamics across the Mid-Pleistocene in the Bering Sea. Simplified schematic of sea ice changes in the Bering Sea across the Mid-Pleistocene showing the sea ice dynamics across the three intervals identified as the early-mid Pleistocene (1.22−1.0 Ma), an interim state (1.0−0.95 Ma), and the mid-to-late Pleistocene (0.95−0.8 Ma and 0.5−0.3 Ma). **a** Sea ice dynamics across glacial intervals of the respective time period. **b** Sea ice dynamics across interglacial intervals of the respective time period. Maps were created using Ocean Data View[69]. Glacial sea level is based on $\delta^{18}O_{seawater}$ from DSDP Site 607[48] and ODP Site 1123[50] with the coastline representing the average minimum sea level across all glacial intervals of that period. The mid-to-late Pleistocene glacial sea ice schematic is additionally supported by studies from the Umnak Plateau[44] and the western Bering Sea[21, 67] across Termination I (blue dots). Continental glacier advances are simplified schematics based on maximum Pleistocene glacier extent in Alaska[70]

during mid-glacials (Fig. 4), whereas the studied glacial periods post ~1.0 Ma show distinct sea ice maxima during the late-glacial/deglacial (Fig. 4), with the exception of MIS 22.

As previously shown by Gildor and Tziperman[17], extensive sea ice cover during glacial terminations negatively impacts snow accumulation on continental glaciers via the temperature–precipitation feedback, limiting evaporation from the polar ocean,

and via the diversion of the winter storm tracks. Model results suggest a sea ice versus land ice hysteresis with a deglacial sea ice spike of 5−10 ka[2], which is of slightly longer duration than the best resolved sea ice maximum during the MIS 25/24 transition (~4.8 ka) recorded at Site U1343 (Fig. 4). Where observed, the transient sea ice maxima are concomitant with deglaciations (Fig. 4), as determined from $\delta^{18}O_{b}$, suggesting that sea ice likely

aids in the initiation of major terminations. Furthermore, the overall increase in sea ice extent across the MPT is in accordance with outcomes from a recent modelling study[18], which indicates that a cooler climate results in larger sea ice extent and an asymmetric sea ice response between hemispheres, leading to 100-ka G/IG cycles[18]. However, since there are, as yet, no Antarctic sea ice records available for the MPT, we cannot directly assess the interhemispheric relationship of sea ice growth. Nevertheless, proxy data across the MPT demonstrates a sea ice/land ice hysteresis as predicted by the SIS hypothesis[1,2,17]. Thus, despite its variable temporal resolution, our sea ice reconstruction from Site U1343 highlights the potential for MPT sea ice change to influence the timing and shape of late Pleistocene climate cycles. This is not only important with respect to understanding long-term sea ice and G/IG dynamics, but also offers the opportunity for improving proxy-model comparisons aimed at assessing the role of sea ice on climate change. Still, further studies from different regions and of even higher resolution are needed, to confirm the presence and duration of sea ice maxima in the Arctic marginal seas across late-glacials/deglacials during late Pleistocene G/IG cycles.

With regard to the SIS hypothesis, we also consider the possible influence of long-term deep ocean cooling on sea ice dynamics[1]. To date, two orbitally resolved bottom water temperature (BWT) records exist across the MPT, one from Deep Sea Drilling Project (DSDP) Site 607 in the North Atlantic[48,49] and another from Ocean Drilling Program (ODP) Site 1123 in the South Pacific[50] (Fig. 4). While both records show divergent trends in BWT history, with no apparent long-term BWT cooling in Site 1123 across the MPT[50], a pronounced decline in BWT from ~1.15 to 0.85 Ma in Site 607[48,49] provides some evidence for a link between deep ocean cooling and sea ice change. The proposed mechanism for sea ice increase as a result of deep ocean cooling is decreased surface ocean heat capacity due to increased stratification[1]. Today, mesoscale eddies form along the eastern Bering slope transporting North Pacific Deep Water (NPDW) to the surface and promoting vertical mixing. Eddy formation is correlated to the strength of the Bering Slope Current[7] (Fig. 1), which in turn is related to Alaskan Stream inflow into the Bering Sea. Decreased Alaskan Stream inflow during glacial intervals from 0.95 Ma[40] could have limited eddy formation, as a result of less vigorous surface ocean circulation, promoting a more stratified water column. Tziperman and Gildor[1] proposed a threshold response of sea ice to deep ocean cooling, which could explain the slight lag between the timing of changes in BWT (1.1 Ma) and sea ice (1.0 Ma). Additionally, the North Atlantic BWT remains low during the late Pleistocene, which could account for the observed increase in sea ice extent during glacial intervals from 0.50 to 0.34 Ma. Yet, more regional, orbitally resolved records of Pleistocene BWT are needed to confirm a long-term deep ocean cooling of the North Pacific and to investigate leads and lags of BWT versus sea ice change in the marginal seas of the Arctic Ocean.

In addition to BWT cooling, changes in sea ice dynamics in the Bering Sea across the MPT are accompanied by regional climate cooling as observed in North Pacific SST records[51] and from regional glacier advances[52]. The increase in sea ice extent in the Bering Sea is consistent with a long-term decrease of North Pacific SSTs (ODP Site 882[51,53], Fig. 4) as a result of the progressive expansion of North Pacific polar water masses from ~1.15 Ma[54]. Long-term North Pacific SST cooling intensified around 1.1 Ma, concomitant with BWT cooling observed at Site 607. Lowest North Pacific SSTs are coincident with the sea ice interim state (1.0–0.95 Ma) (Fig. 4). Brunelle et al.[55] argue that low mean ocean temperatures during glacial intervals lead to a decreased temperature sensitivity and increased salinity stratification in polar and sub-polar regions. A simultaneous decrease in

North Pacific SSTs, recorded in Site 882[51], and North Atlantic BWTs (Site 607[48,49]) from ~1.1 Ma would support homogenous cooling and increased salinity stratification during glacial periods promoting sea ice formation as a result of decreased heat capacity of the surface ocean. An increase in North Pacific SSTs between 0.95 and 0.85 Ma has been attributed to a northward movement of the North Pacific Polar Front[54]. However, MIS 12 and 10 show increased sea ice extent (Fig. 4) even though SSTs from Site 882 remain high from ~0.85 Ma onwards and increase even further from ~0.5 Ma (Fig. 4)[51]. This indicates that the northward shift of the North Pacific Polar Front did not propagate into the Bering Sea as sea ice duration and extent increase throughout the late Pleistocene interval. The increase in sea ice extent in the Bering Sea thus paralleled North Pacific SST and North Atlantic BWT decrease, together with regional glacier advances, suggesting a response to global climate cooling.

Finally, we investigate the importance of sea ice extent in the eastern Bering Sea for GNPIW formation. The difference between benthic foraminiferal carbon isotope records ($\delta^{13}C_b$) recorded at ODP Site 849[56] (0°11.0′N, 110°31.1′W; 3851 m) and IODP Site U1342 (54°49.7′N, 176°55.0′E, 818 m), $\Delta\delta^{13}C_{(849-U1342)}$, is used as a proxy for Bering Sea GNPIW formation/ventilation as proposed by Knudson et al.[10] (Fig. 4). Specifically, since the Site 849 $\delta^{13}C_b$ is believed to approximate global oceanic dissolved inorganic carbon ($\delta^{13}C_{DIC}$) values[56], subtraction of site-specific $\delta^{13}C_b$ values enables local influences to be determined[10]. Site U1342 is located in the southern Bering Sea on the Bowers Ridge (Fig. 1) and lies just below the modern depth of NPIW (300–800 m[57]) at 818 m water depth. Currently, Site U1342 is bathed in NPDW with very low $\delta^{13}C_{DIC}$[10]. During glacial periods, however, GNPIW was formed in the Bering Sea and reached down to the seafloor (at U1342), transporting high $\delta^{13}C_{DIC}$ surface waters to greater depth[10]. Thus high (low) $\Delta\delta^{13}C_{(849-U1342)}$ represents reduced (enhanced) GNPIW influence at Site U1342 (Fig. 4). U1343 IP$_{25}$ and $\Delta\delta^{13}C_{(849-U1342)}$ are inversely correlated (Supplementary Fig. 3), supporting a role of sea ice for GNPIW formation via brine rejection, which is consistent with previous studies from the western and eastern Bering Sea, indicating intermediate water formation[8,9,57].

Increased sea ice cover during glacial periods also has important implications for ocean-atmosphere gas exchange. Presently, deep reaching eddies bring high-$CO_2$ NPDW to the surface ocean along the eastern Bering slope[7], resulting in occasional $CO_2$ outgassing[58]. It is possible that increased sea ice extent in the Bering Sea during late and post-MPT glacial periods aided glacial atmospheric $CO_2$ drawdown via two mechanisms. First, by carbon sequestration during the formation of GNPIW, and second, by reducing $CO_2$ outgassing via decoupling of the deep and surface ocean and by acting as a physical barrier. Increased sea ice extent in the Bering Sea across the Mid-Pleistocene thus has the potential to aid increased glacial abyssal North Pacific carbon storage, as proposed by studies from Site 882 during (at least) the late Pleistocene[59]. Future work should focus on reconstructing the depth of GNPIW across the late Pleistocene and its ability for carbon sequestration, as this would likely have been influenced by the amount of brine formation and thus sea ice dynamics. Additionally, future multi-proxy reconstructions of past sea ice variability in the marginal seas of the North Pacific should be coupled with investigations of GNPIW ventilation and carbon biogeochemistry as such studies could help unravel the role of sea ice for deep ocean carbon storage.

In summary, our sea ice reconstruction from Site U1343 in the Bering Sea shows a twofold change across the MPT (Fig. 5) with important implications for the SIS hypothesis, GNPIW formation, and potentially glacial North Pacific carbon storage. An increase in sea ice extent from ~1.15 Ma, likely a result of regional

and global climate cooling, was accompanied by the occurrence of a consistent late-glacial/deglacial sea ice maximum in all but one studied glacial periods post ~1.0 Ma (Figs. 4 and 5), as predicted by the SIS hypothesis[1,17]. This study is an important step forward in understanding the role of sea ice for the MPT, but will need to be supplemented by further high-resolution (and continuous) orbitally resolved sea ice records over the past 1.5 Ma to determine the exact timing of sea ice build-up across G/IG cycles. Additionally, sea ice records from other Arctic marginal seas across the MPT should help confirm the occurrence of deglacial sea ice maxima in late-to-post MPT glacial periods and to further understand their implications. Finally, we note that additional studies of Bering Sea surface ocean chemistry are needed to determine the role of sea ice for $CO_2$ sequestration and ultimately deep ocean carbon storage via GNPIW formation.

## Methods

**Regional settings and chronology**. IODP sediment core U1343 (57°33.4′N, 176° 49.0′W, water depth 1950 m) was retrieved off the eastern Bering Sea continental margin (Fig. 1) on a topographic high, to reduce the impact downslope transport[41]. At present day it is bathed in NPDW that enters the Bering Sea through several deep passes in the Aleutian Island Arc[60]. The surface circulation in the Bering Sea forms a cyclonic gyre (Fig. 1). Surface water enters the Bering Sea through the Aleutian passes and main surface outflow occurs through Kamchatka Strait in the west (Fig. 1). Some surface water (0.85 Sv[61]) flows northward and leaves the Bering Sea through the 50 m deep Bering Strait—the gateway between the Pacific and Arctic Ocean. Mesoscale eddies developing in the eastern Aleutian basin bring nutrient-rich waters to the surface and maintain high primary productivity[7] in the eastern Bering Sea. Other shelf edge processes such as tidal mixing and transverse circulation further enhance the horizontal exchange of nitrate-rich basin waters and iron-rich shelf waters[6] sustaining high primary productivity (175–275 g C m$^{-2}$ yr$^{-1}$; ref. [6]) along the continental margin, also called the 'Green Belt'. The eastern Bering Sea shelf is characterised by an intensive spring phytoplankton bloom, as a result of nutrient release during sea ice melting[5,62]. In recent decades the Bering Sea has experienced substantial retreat in the winter sea ice margin and earlier sea ice melting in spring with important implications for the marine ecosystem[63], demonstrating the sensitivity of sea ice to climate change. The combination of Site U1343 being close to the present day winter sea ice margin (Fig. 1) and the recent sea ice decrease in the Bering Sea makes it an ideal location to study past sea ice extent and to understand past sea ice dynamics in changing climates.

In total five holes were drilled at Site U1343 (A–E) out of which three (A, C, E) were used to construct a composite depth scale (splice) based on the physical properties of the cores from 0 to 269.92 m CCSF-A. Below the splice, U1343E cores with unknown gaps are appended from ~270 to 779 m CCSF-A[42]. The age model of U1343 is based on oxygen isotope stratigraphy by correlating the continuous oxygen isotope record to the global LR04 stack[16,42]. Oxygen isotope measurements at Site U1343 are based on seven benthic foraminifera species (*Cibicidoides* spp., *Elphidium batialis*, *Globobulimina pacifica*, *Nonionella labradorica*, *Planulina wuellerstorfi*, *Uvigerina bifurcata*, and *Uvigerina senticosa*), and normalised to *E. batialis*, the most abundant species at Site U1343[42]. Due to sample resolution this approach yields a highly refined age model for the last 1.2 Ma and a refined age model between 1.2 and 2.4 Ma[42]. The age model based on oxygen isotope stratigraphy is in good agreement with datum events based on bio- and magnetostratigraphy[42].

**Extraction and analysis of biomarkers**. HBI lipids were extracted from 3 g of freeze dried homogenised sediments as described in Belt et al.[64]. Additionally, removal of elemental sulphur was performed[65] and sterol fractions were collected. Prior to analysis samples were kept in cold storage at 7 °C. Samples were freeze dried at −45 °C and 0.2 mbar for 48 h using a Thermo Savant Modulyo D freeze drier and an Edwards K4 Modulyo freeze drier at Plymouth and Cardiff University, respectively. After freeze drying, samples were homogenised using a dichloromethane (DCM)-cleaned agate pestle and mortar and 3 g of sediment was weighed into 7 ml glass vials with aluminium-lined polypropylene screw caps. In addition to the sediment samples, a blank and two samples of standard sediments with known biomarker concentrations were added to each extraction batch. Standard sediments are from the Canadian Arctic Archipelago and 0.5 g of sediment were weighed in per sample. Ten microlitres of 0.01 mg mL$^{-1}$ 9-octylheptadec-8-ene (9-OHD) and 5 α-androstan-3βol solution were added to each sediment vial and procedural blank as internal standards for HBI and sterol quantification, respectively. The samples were extracted three times using a mixture of DCM (high performance liquid chromatography (HPLC) grade) and methanol (MeOH, HPLC grade) 2:1 (v/v). Gas chromatography of the first extraction batch showed high concentrations of elemental sulphur in Site U1343 sediment samples that interfere with IP$_{25}$ analysis. Therefore sulphur removal[65] using tetrabutylammonium sulphite reagent was performed for all samples prior to silica column chromatographic purification of the total organic extracts (TOE)[64]. During silica chromatography, non-polar components (HBIs, including IP$_{25}$) were eluted with hexane and collected in new pre-labelled 7 mL glass vials (TOE-2), whereas more polar hydrocarbon fractions (sterols) were eluted using hexane/methyl acetate (1:4, v/v) and collected in separate pre-labelled 7 mL glass vials (TOE-3). TOE-2 was dried under N$_2$ flow at 25 °C and re-dissolved in 150 μL hexane before being transferred to 300 μL GC glass vials, concentrated to 20 μL under N$_2$ flow at 25 °C, and capped with aluminium crimp-top caps and Teflon septa (Chromacol Ltd., UK). Due to low abundance of HBI lipids and high concentrations of n-alkanes HBI fractions from the oldest time interval (1.53–1.36 Ma) were additionally purified using silver-ion chromatography (5:95 AgNO$_3$:SiO$_2$) to remove n-alkanes. TOE-3 was dried under N$_2$ flow at 25 °C and derivatised using N,O-Bis (trimethylsilyl)trifluoroacetamide (50 μL, 70 °C; 1 h). All samples were analysed by gas chromatography-mass spectrometry (Agilent 7890A GC coupled to a 5975 series mass selective detector fitted with an Agilent HP-5ms column) at Plymouth University using the operating conditions specified in Belt et al.[64]. The identification of individual lipids was based on their characteristic retention times and mass spectra and quantification was achieved by integrating the peak area of selected ions (m/z 350 (IP$_{25}$); 346 (HBI III); 470 (brassicasterol)) in comparison to the peak area of the internal standards added to each sample[64]. Quantification of individual lipids also considers an instrumental response factor obtained from known concentrations of biomarker lipids in the standard sediments[64].

**Defining sea ice boundary conditions**. In order to reconstruct different sea ice states, we use a combinatory approach of IP$_{25}$ (indication of seasonal sea ice), HBI III (most prominent in the MIZ), and the mass accumulation rate of biogenic opal (MAR$_{opal}$, first-order changes in export productivity)[36], measured in IODP Site U1343 in the eastern Bering Sea. IP$_{25}$ and U1343 MAR$_{opal}$ are weakly anti-correlated (IP$_{25}$-MAR$_{opal}$: $r_{xy} = -0.235$ with 95% Student's $t$ confidence intervals [−0.420; −0.031], $n = 142$), consistent with the interpretation, that an increased sea ice cover leads to decreased primary productivity as a result of light limitation in the surface ocean.

Here, we build on the approach outlined in Méheust et al.[21], who used threshold values for IP$_{25}$ and sedimentary biogenic opal content in the western Bering Sea to identify sea ice regime shifts over the past 20 ka BP. Site U1343 is located on a topographic high off the eastern Bering slope, which reduces the influence of downslope transport. However, to avoid potential siliciclastic dilution effects, we used the MAR$_{opal}$[36], rather than the sedimentary biogenic opal content (wt. %). Here we developed a multi-proxy classification, where threshold values of IP$_{25}$, HBI III and the MAR$_{opal}$ are used to identify four different sea ice states (Table 2). The threshold values of IP$_{25}$ and HBI III represent 10% of the total range of the respective biomarker (total range of IP$_{25}$: 0–5 ng g$^{-1}$ sed, total range of HBI III: 0–7 ng g$^{-1}$ sed) and are summarised in Fig. 4, Supplementary Fig. 1, and Supplementary Fig. 4, while the corresponding value in the MAR$_{opal}$ is based on the past 20 ka. Sea ice reconstructions using IP$_{25}$ and sea ice diatoms from the western Bering Sea[21] and the Umnak Plateau[44], respectively, indicate a change from a more extended sea ice cover to seasonal sea ice/ice-free conditions around 15 ka BP across Termination I, suggesting that this is a common feature of sea ice dynamics in the Bering Sea. The MAR$_{opal}$ value in Site U1343 at 15 ka is 4 g cm$^{-2}$ ka$^{-1}$ (Supplementary Fig. 5), which is used as the threshold value to distinguish between an extended sea ice cover and more seasonal sea ice/ice free conditions (Table 2). In contrast to the western Bering Sea[21], seasonal sea ice conditions in the eastern Bering Sea (as indicated by increased IP$_{25}$) are characterised by variable MAR$_{opal}$ values. This could be a result of the dynamic high productivity region overlying the core Site of U1343, indicating that sea ice concentration, even though of major importance, might not be the only influence on primary productivity along the eastern Bering slope. Only three data points do not fit with our overall sea ice state classification (Table 1), as they have low IP$_{25}$, but HBI III values outside of our threshold for the extended/ice-free states. The reasons for this are unclear, but potentially indicate predominantly ice-free/perennial sea ice conditions with occasional MIZ sedimentation. For these data points, the MAR$_{opal}$ can be used to distinguish between ice-free and extended ice cover scenarios (Table 1, Table 2). The outcomes of the sea ice state classification, as applied to samples from Site U1343 are illustrated in Supplementary Figs. 1 and 4.

**Statistical analysis**. For calculating Pearson's $r$ correlation coefficients, the software package PearsonT3[66] was used. PearsonT3 automatically performs mean detrending of the data and estimates the persistence time of both variables. Persistence is a common feature in climate records; however, it reduces the effective data size. As the data size ($n$) is small to begin with, confidence intervals are quite large. Ninety-five per cent confidence intervals are estimated using equi-tailed bootstrapping and are of the type Student's $t$.

**Data availability**. All data generated during this study supporting its findings are supplied via the NERC Polar Data Centre (UK-PDC), doi:10.5285/9caf74c4-7054-4539-81b8-d4f942afc358.

# ARTICLE

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

## Acknowledgements

H.D. would like to acknowledge funding through a Natural Environmental Research Council (NERC) Ph.D. research grant (NE/L002434/1), via the GW4+ Doctoral Training Partnership and additional funding provided by a BGS University Funding Initiative Ph. D. studentship (S268). Marine sediment samples were provided as part of the International Ocean Discovery Program (IODP). S.T.B., L.S., and P.C.-S. thank the University of Plymouth for funding to support biomarker analysis.

## Author contributions

H.D., S.M.S., and S.T.B. developed this sea ice study using Arctic biomarkers. K.H. and S. K. sailed on IODP Leg 323 and contributed to developing the MPT study at Site U1343, with valuable sample material provided by K.H. H.D. performed most of the analyses with support from L.S. and P.C.-S. H.D. wrote the manuscript with help from S.T.B. and S.M.S. and contributions from L.S., K.H., S.K., C.H.L., and I.R.H.

## Additional information

**Competing interests:** The authors declare no competing financial interests.

