## [Peer Review File · Nature Communications]

Reviewers' comments:

Reviewer #1 (Remarks to the Author):

Review of "Changes in eastern Bering Sea sea ice dynamics across the Mid-Pleistocene climate transition" by Detlef et al, Nature Communications.

This is an exciting paper, presenting a first evidence for the timing of sea ice changes vs glacial terminations during ice ages, and shedding light on an important long standing climate problem. The work uses a novel sea ice biomarker in conjunction with other biomarkers to deduced the seasonal extent of sea ice over a critical period in Earth climate history, from 1.5 Myr (million years) ago to about 0.3 Myr, near the Bering Straights. This time interval includes the time, about 0.8 Myr ago, in which glacial cycles dramatically changed from a period of 41 kyr (41,000 years) to the current 100 kyr cycles. We believe we understand the 41 kyr cycle to be most likely a direct response to Milankovitch forcing, yet the mechanism of the 100 kyr cycles and of the "mid-Pleistocene transition" (MPT) between the two regimes is a longstanding problem in climate science. This ignorance is especially embarrassing given that glacial cycles are the strongest climate variability signal over the past few millions of years.

It is commonly believed that sea ice plays an important role in climate, although its suggested role in glacial cycle dynamics specifically has been speculative so far. The current paper makes a very significant contribution, in assessing for the first time the sea ice extent as function of time during the glacial cycles, before and after the mid-Pleistocene transition. This is a major improvement relative to previously available sea ice extent estimates during the last glacial maximum, which provide a spatial map, but not temporal resolution. The current paper shows that sea ice extent generally increases over a long time scale and across the MPT, and appropriately explains that this is a result of other processes. This is an important result by itself. But I find the findings of the specific phase of sea ice extent increases and decreases during the 41 kyr cycles vs the 100 kyr cycles to be even more remarkable. Specifically, the authors find that "sea ice extent from 1 Ma onwards peaked during late glacials/early terminations", or in other words, that sea ice is maximal just prior to and during the beginning of terminations of the 100 kyr ice ages. This is a remarkable, novel and seemingly non-intuitive observational result, even if consistent with the sea ice switch (SIS) idea as explained by the authors, and therefore a major contribution. In general, I find the discussion of previous theories, including the SIS, to help put the results in context in a way that makes them of interest to a large community of climate scientists, whether dynamicists, modelers or geochemist dealing with glacial cycle observations.

Addressing the general review criteria of Nature Communications, I would note that the data are technically sound, and in fact rely on a novel proxy; the paper certainly provides strong evidence for its conclusions, as well as a careful discussion of limitations and caveats; some of the discussion is necessarily speculative (role of different water masses in setting sea ice extent, for example), yet when that is the case the paper is careful to note the necessary caveats; the results are certainly novel and even exciting. And finally, it is my feeling that the manuscript is important to scientists in more than a specific field.

I am therefore happy to recommend the publication of this paper in Nature communications.

Reviewer #2 (Remarks to the Author):

Ms Detlef and co-workers present new biogeochemical datasets generated on three intervals including pre, during and post Mid-Pleistocene Transition (MPT) from downcore sediments deposited in the Bering Sea. Their work tries to illustrate the relationship between changes in the MPT sea ice cover dynamics –as reconstructed at IODP Site U1343– and late Pleistocene climate

cycles. The Bering Sea -and the subarctic Pacific more generally- is considered to have played a central role in past carbon sequestration in the deep sea with implications for interglacial-glacial (IG-G) changes in Pleistocene climate. The authors propose that a twofold change toward increased seasonal sea ice duration and extent across the MPT and the evolution of a deglacial sea ice maximum were essential to define late Pleistocene climate cycles.

We are currently overly reliant on ODP Site 882 for our understanding of high latitude processes in the Subarctic Pacific. In this regard, Ms Detlef's work will become an important contribution to the state of art for the high latitude North Pacific. Having read their Ms, I therefore find myself asking 'how regionally representative is the sea ice regime in the Bering Sea for the subarctic Pacific'. Answering this question with respect to CO₂ drawdown across the MPT seems to me to be the bigger picture. For me, their principal conclusion - that sea ice cover in the Bering Sea experienced a twofold development and that the drivers of such condition seemed to have evolved during a short time window during the MPT - is insufficiently supported by their Results. Previous modelling studies suggest that changes in sea ice dynamics may have been responsible for the shift of prolonged G/IG cycles; this hypothesis is tested by Detlef and colleagues with sea ice proxy data. I am, however, currently left somewhat dissatisfied by the end-product.

Several issues and questions are raised below:

Introduction

Curiously enough, the pioneer work at IODP Site 882 is hardly mentioned. Site 882 has become the reference and comparison location for any Plio-Pleistocene study dealing with paleoclimate and paleoceanography from the high-latitude North Pacific.

The present-day setting in the study area and surroundings is insufficiently described. An audience less familiar with climate, oceanography and ice cover dynamics of the Arctic and Subarctic Pacific would hardly understand the physical background behind this paleostudy.

There are some methodological issues the authors introduce (see l. 71), but are hardly considered afterward in the Discussion.

Results

l. 108-109: although it is true that 'a reduced IP25 concentration is consistent with low abundances of sea ice associated diatoms in U1343 reported previously', it is also true that sea ice diatoms identified at U1343 (Teraishi et al., 2016, DSR II 125-126: 18-28) are not those producing IP25 (*Haslea kjellmanii*, *Haslea crucigeroides* and *Pleurosigma stuxbergii*). How do the authors explain this mismatch?

l. 121: What do the authors mean with the 'asymmetry' of G/IG cycles? Differences in timing and amplitudes? With 'slow glaciations' the authors mean slow start of a glaciation?

l. 127-129: The authors state that 'increase in peak glacial IP25 values is observed from MIS 28 onwards concomitant with a shift in the timing of the glacial IP25 maximum from mid-glacial to late glacial/early termination', though this change is not reflected by the U1343 d18O signal. Why not?

l. 130: the increase in 'late interglacial IP25 values' is only seen in the transition between MIS 29 and 28 and it is a feature not repeated in later G/IG transitions.

l. 131: why do the authors state that the 'concentration of HBI III is relatively low prior to MIS 31', when peak values are around 150 ng g⁻¹ lower prior to MIS 29 compared to later?

l. 155: 'during interglacial intervals of MIS 13 and 10'. This is just wrong: (i) MIS 13 is an interglacial, no need to write 'interglacial intervals of MIS 13', which implies that MIS 13 could also have glacial intervals; (ii) MIS 10 is a glacial.

l. 162-163: this is one peak recorded in only one sample (sample resolution 10 ky). Therefore I would be careful in interpreting this as a 'momentary return to seasonal sea ice cover'.

Discussion

The authors tend to over-interpret and draw to wide-scoping conclusions without offering convincing and strong arguments supporting their interpretations. The first sentence of this chapter speaks for the whole Ms: it is argued that 'variations of sea ice and open water biomarkers during the past 1.5 Ma demonstrate major changes in NH sea ice dynamics'. Their non-continuous analysis focus on three separated intervals and interpretations cannot be reliably extrapolated to the past 1.5 Ma.

Several issues are shortly introduced without any further discussion. An example of this is the short explanation on the decoupling between Brassicasterol, and HBI III and IP25 across the interim state (Fig. 3).

I. 219: it is incorrect to state that the low values of IP25 during MIS 22 'may reflect the relatively low sample resolution, especially across the termination.' Actually eight samples are assigned to MIS 22, a higher amount than that of previous G.

I. 220: I do not understand why the authors state 'that the inverse relationship between IP25 and HBI III (Fig. 3) throughout the late Pleistocene indicates the renewed onset of pronounced seasonal sea ice cyclicity in the eastern Bering Sea'. Figure 3 presents results of the MPT, not temporally located in the late Pleistocene.

I. 227: since there is a 300 ka-gap in the dataset between late MPT and MIS 13, it is misleading to state that 'MIS 12 is the first glacial throughout the entire record where perennial sea ice is encountered'. A few lines below is stated that 'perennial sea ice conditions are also recorded at the end of MIS 10 suggesting that this could be a common feature of past MPT climate cycles'. How can this be stated when the studied post-MPT interval is limited to a narrow time window of less than 200 ka (around the Mid-Brunhes Event) and an in-between 300 ka-gap exists?

I. 247: Terminations should be clearly identified in Figs. 2 and 4.

I. 248-250: This kind of statements (potential role of sea ice in increasing atmospheric CO₂ concentrations or change in sea ice dynamics as a consequence of Mid-Brunhes Event) is too vague.

I. 259: how did the authors determine that the peak during late MIS 26 had a duration of 4 ka, when sample resolution is 5 ka for the MPT? (see I. 118).

I. 262-280: it is true that the long-term cooling intensified around 1.1 Ma at Site 882 (roughly matching the interim state at Site 1343) and that the U1343 diatom record shows growing influence of the warm and salty Alaskan Stream into the interim state. However, SSTs at site 882 became warmer afterward, while the relative contribution of *Neodenticula seminae* (proxy for the Alaskan Stream) remained low. How this match the proposed 'renewed seasonality of sea ice' (I. 275)?

I. 295-300: The authors give the reader the impression that they are about to do more than just highlighting the potential of the Bering Sea to play a role in CO₂ drawdown during the MPT and afterward. Of course, the lack of a CO₂ record did not stop those working on Site 882 from speculating on the role of the Subarctic Ocean in carbon sequestration across the onset of NHG 2.7 Ma, but it is necessary to remember that we still do not know what happened to CO₂ across the MPT. Whatever happened to polar ocean CO₂ sequestration (see also I. 248-250), it is unlikely that have contributed to the origin of the MPT. The authors shortly address the link between CO₂ and ice free conditions, but this contributes neither to the main Ms message nor to solve the question on ice ages and CO₂ content in the atmosphere.

The main message of the Discussion subsection 3.3 seems to be the change in sea ice cover during the MPT was essential for the timing and behavior of late Pleistocene climate cycles. It does not add much new to the general discussion and does not offer a clear and focused 'take-home' message for the reader nor clearly present future perspectives in paleoclimatic reconstruction. They state that their 'data support the influence of sea ice on the timing and shape of late Pleistocene climate cycles' (I. 324-25). The end of Ms leaves the reader even more confused: a scenario of increase in sea ice extend, SST, BWT and regional glacier advances is postulated as a response to global climate cooling (I. 333-334).

Figure 2: there are several issues with this figure.

- I assume that light blue and light green(?) lines in 2c (IP25 and HBIII) represent the lower concentrations of IP25 and HBIII compared to those in 2B and 2A: this is not explained in the caption.

- The $\delta^{18}O$ -based stratigraphy in 2c is poorly resolved. The ups and downs hardly allow distinguishing between G and IG.

- Why is LR04 Stack shown in 2C, but it is absent in 2B and 2A?

- 2A: the caption 'Marine isotope stage (MIS) 13 to 10 spanning the Mid Brunhes event' is misleading: the Mid-Brunhes event roughly occurred around 430 ka, matching the transition

between MIS 12 and 11. It should be clearly identified in this figure.

- Why having two different y-axes for IP25 and HBI III in 2C? It is ok if concentrations are lower than in 2B and 2A, but having two axes for the same proxy with two different scales in the figure is unnecessary and confusing.

- 2B: why MPT is called here 'Mid-Pleistocene CLIMATE Transition' while is Mid-Pleistocene Transition throughout the MS?

Reviewer #3 (Remarks to the Author):

The authors present biomarker records in a sediment core from the eastern Bering Sea for sea-ice reconstruction to understand the Mid-Pleistocene transition (MPT) mechanism. The sea ice biomarker (IP25) and phytoplankton biomarkers (brassicasterol and HBI III) are combined with published $\delta^{18}O$ data, as well as compared with the regional and global climate records across the MPT. For the first time, the IP25 sea-ice biomarker record is used for the MPT in the North Pacific. The main finding provided by Detlef et al. is a twofold increased sea ice concomitant with a deglacial sea ice maximum across the MPT.

The results are interesting and potentially worth published in nature communication, but it is difficult to follow the interpretation of the biomarker data sometimes and there is a loose link to the displayed records. I recommend major revision for this manuscript.

Introduction

-The objects are missing in the introduction.

-Lines 58-86: in this paragraph, the core location setting (lines 62-66) and the introduction of biomarkers (lines 67-77) need to be restructured. The start of the biomarker introduction is kind of incoherent.

-Lines 75-83: Belt et al. (2015, EPSL) and Smik et al. (2016, OG) demonstrated that the marginal ice zone is characterized with the elevated HBI III contents in the Barents and Norwegian Sea using surface-sediment calibration. However, HBI III is abundant in surface sediments from the North Iceland Shelf influenced by the drift ice (Cabedo-Sanz et al., 2016, QSR). Therefore, the authors should be careful with their interpretation due to the lack of surface-sediment calibration in the Bering Sea. This is very important for this manuscript.

Results

-Lines 97-98: to my knowledge, absent IP25 along with low phytoplankton biomarkers should suggest perennial sea-ice condition. The wrong interpretation of biomarker data can mislead the discussion derived from the data set.

-Lines 105-108: IP25 and HBI III concentrations are generally during MIS 51-44 while brassicasterol concentrations are comparable to those during MIS 36-20 and MIS 13-10. The authors interpreted this as the degradation in this older section, but IP25 should be stable than brassicasterol preserved in marine sediments.

-Lines 162-164: Why the HBI III concentration is still low during this momentary return of seasonal sea ice?

Discussion

-Lines 180-182 : the low/absent IP25 in Arctic sediments is considered to reflect either ice free conditions or permanent ice cover as the authors mention in the introduction section. The predominantly ice-free conditions proposed by the authors here should be interpreted with the phytoplankton biomarker.

-Lines 204-206: actually, the brassicasterol has a variety of sources, including marine and riverine sources, and even sea ice origin. Thus, the extended sea ice cover can not necessarily result in low brassicasterol concentrations. The brassicasterol concentrations in surface sediments from the Barents and Norwegian Sea are also abundant under the marginal ice zone as the same as HBI III (Belt et al., 2015, EPSL).

-Lines 227-229: I only see low HBI III concentrations of MIS 12, but the concentrations of IP25

and brassicasterol are variable. Cite references to support this assertion.

-Concerning the mechanisms of sea ice changes, the authors propose that an extensive sea ice cover causes the land glacier retreat via the temperature-precipitation mechanism. However, the response of sea ice to the climate change should also be taken into count.

Figure

-Figure 4: the curves indicating IP25 and HBI III should be labelled.

-Although Figure 4 has been cited in the discussion section several times, the interpretation is rather loose to the figure.

Replies to reviewer's #1 comments:

Thank you very much for your altogether supportive review of our manuscript, we appreciate the time you have put in to reading our work and thank you for recognizing the novelty of our dataset.

Replies to reviewer's #2 comments:

- 1. In this regard, Ms Detlef's work will become an important contribution to the state of art for the high latitude North Pacific.**

Thank you very much for acknowledging the novelty of our research, especially with regard to this highly 'under-sampled' region with regard to the Mid-Pleistocene.

- 2. 'how regionally representative is the sea ice regime in the Bering Sea for the subarctic Pacific'**

The North Pacific sea ice regime is mainly characterized by seasonal sea ice formation in the Bering Sea and the Sea of Okhotsk. On glacial/interglacial timescales both regions play an important role with respect to North Pacific Intermediate Water (NPIW) formation as a result of brine rejection, during sea ice freezing. At present day NPIW forms in the Sea of Okhotsk, however studies show that NPIW was at least partly formed in the Bering Sea during glacial intervals (Rella et al. 2012, Knudson et al. 2015, Max et al. 2017). Ohkushi et al. 2003 even propose a source switch for NPIW during warm periods (Sea of Okhotsk) and cold periods (Bering Sea). However, we cannot rule out completely the formation of NPIW in the Sea of Okhotsk during Pleistocene glacials, as to date no records exist. Nevertheless, our data clearly support the influence of eastern Bering Sea sea ice formation on NPIW formation, as proposed by Rella et al. 2012. Further the Bering Sea forms the gateway between the Arctic and Pacific Ocean and thus, at times of an open Bering Strait, sea ice extent and resultant oceanographic changes, such as brine rejection, nutrient release during spring ice melting or nutrient uptake during spring phytoplankton blooms will have direct consequences for the Arctic Ocean via Bering Strait outflow. Additionally, we think that Bering Sea sea ice extent is more likely to impact the North American Ice Sheets as air masses reaching North America may be more influenced by central and eastern subarctic North Pacific water. We thus argue that the Bering Sea is representative for the North Pacific sea ice regime, especially with respect to the purpose of this study, investigating the timing of sea ice mainly across Pleistocene glacial intervals. To fully address the reviewers comment we added a specific section on the North Pacific sea ice regime to the introduction (l. 6-15).

- 3. Answering this question with respect to CO₂ drawdown across the MPT seems to me to be the bigger picture**

We very much appreciate this insightful feedback, and have emphasised the potential contribution of sea ice for glacial atmospheric CO₂ drawdown (l. 279-286). To aid our interpretations we added a record of $\Delta\delta^{13}\text{C}_{(\text{ODP } 849- \text{IODP } \text{U1342})}$ from the southern Bering Sea

(Knudson et al. 2015) (Fig. 4), thought to reflect Bering Sea intermediate water ventilation and thus representing qualitative changes in North Pacific Intermediate Water (NPIW) formation. The negative correlation of IP_{25} and $\Delta\delta^{13}C_{(ODP\ 849- IODP\ U1342)}$ (supplementary Fig. 2) significantly demonstrates the importance of eastern Bering Sea sea ice extent for NPIW formation during glacials, most likely as a result of brine rejection. Increased NPIW production has the potential to aid in abyssal North Pacific carbon storage.

- 4. Curiously enough, the pioneer work at IODP Site 882 is hardly mentioned. Site 882 has become the reference and comparison location for any Plio-Pleistocene study dealing with paleoclimate and paleoceanography from the high-latitude North Pacific.**

We acknowledge that ODP Site 882 has become an important site for Plio-Pleistocene research in the North Pacific. However, studies have mainly focused on the onset of northern hemisphere glaciation (Haug et al. 2005), during the Pliocene, and more recent time periods, such as the late Pleistocene (Jaccard et al. 2005), or the last glacial maximum (John & Kriesek, 1990 (IRD)). Further we do compare our data to sea surface temperature records from ODP 882 (Martinez-Garcia et al. 2010), that correlate with sea ice increase across the Mid-Pleistocene transition (Fig. 4). In the context of North Pacific abyssal carbon storage we also reference Jaccard et al. 2009, a record of benthic foraminiferal Cd/Ca ratios over the past 150 ka from ODP Site 882.

- 5. The present-day setting in the study area and surroundings is insufficiently described.**

We acknowledge this was the case. Following up from question no. 2 and 5 we added a paragraph to the Introduction on the present day sea ice regime in the Bering Sea and its implications for the North Pacific (l. 6-15). Current systems and patterns of primary productivity in the eastern Bering Sea can be found in the method section 'Regional settings and chronology' (l. 300-315).

- 6. There are some methodological issues the authors introduce (see l. 71), but are hardly considered afterward in the Discussion.**

We acknowledge that the original description of the biomarker proxies was unwittingly misleading. We now provide a more focused explanation to equip the reader with key background knowledge, which we believe is pertinent for the interpretation (l. 43-63). Namely the occurrence of IP_{25} as an indicator of seasonal sea ice, the use of HBI III as a phytoplankton biomarker, and the phase behaviour of the two. We have decided to not interpret brassicasterol in terms of the changing sea ice regime, as recent studies have demonstrated the ambiguity of brassicasterol, which can have marine, lacustrine, and sea ice sources (for further explanation please see answer to question no. 4, reviewer #3).

7. **I. 108-109: although it is true that ‘a reduced IP₂₅ concentration is consistent with low abundances of sea ice associated diatoms in U1343 reported previously’, it is also true that sea ice diatoms identified at U1343 (Teraishi et al., 2016, DSR II 125-126: 18-28) are not those producing IP₂₅ (Haslea kjellmanii, Haslea crucigeroides and Pleurosigma stuxbergii). How do the authors explain this mismatch?**

We thank the reviewer for raising this good point. As shown by Brown et al (2014), the IP₂₅ producers constitute a reasonably consistent proportion of the total assemblage of sea ice diatoms (Brown et al. 2014 give a value of 1–5%). As such, IP₂₅ (a measure of typically 1-5% of sea ice diatoms) content should reflect the variability in total sea ice diatom content, which should both be driven by sea ice.

8. **I. 121: What do the authors mean with the ‘asymmetry’ of G/IG cycles? Differences in timing and amplitudes? With ‘slow glaciations’ the authors mean slow start of a glaciation?**

With asymmetry of glacial/interglacial cycles we aimed to draw attention to the increase in the ‘sawtooth’ profile of glacial/interglacial cycles across the Pleistocene (slow onset, rapid termination). However, on further reflection, we decided that this does not add to the main point of the manuscript, and we removed all the references to the asymmetry of glacial/interglacial cycles.

9. **I. 127-129: The authors state that ‘increase in peak glacial IP25 values is observed from MIS 28 onwards concomitant with a shift in the timing of the glacial IP25 maximum from mid-glacial to late glacial/early termination’, though this change is not reflected by the U1343 d18O signal. Why not?**

We thank the reviewer and acknowledge this valid point. We argue in the manuscript that sea ice increase in the Bering Sea across the Mid-Pleistocene is a result of regional and global climate cooling, especially with respect to expansion of polar water masses seen in decreased North Pacific sea surface temperatures and regional glacier advances on the North American continent. However we cannot rule out the influence of deep ocean cooling on sea ice increase across the MPT, as proposed by the SIS. The $\delta^{18}\text{O}_b$ record of Site U1343 reflects bottom water temperatures, continental ice volume, and potential changes in salinity both on a regional and local scale. To date no bottom water temperature record exists, so we are unable to deconvolve the record into its temperature and ice volume component. Thus we think the increase in sea ice extent and the shift to a late glacial sea ice peak does not have to be reflected in the $\delta^{18}\text{O}_b$ record of U1343.

10. **I. 130: the increase in ‘late interglacial IP25 values’ is only seen in the transition between MIS 29 and 28 and it is a feature not repeated in later G/IG transitions.**

We thank the reviewer for the comment but after reviewing the data again an increase in late interglacial IP₂₅ values can also be observed during MIS 27, MIS 25, and MIS 21 (Fig. 2).

However, on reflection such observations add little to the main message of the manuscript so we've taken out this short sentence.

11. l. 131: why do the authors state that the 'concentration of HBI III is relatively low prior to MIS 31', when peak values are around 150 ng g⁻¹ lower prior to MIS 29 compared to later?

We thank the reviewer for the comments but we don't necessarily follow it. When reviewing the data HBI III is never as high as 150 ng g⁻¹ sed. From our perspective peak HBI III values prior to MIS 31 vary between 0-2 ng g⁻¹ sed. During late MIS 31/early MIS 30, however peak values increase and vary between 2-8 ng g⁻¹ sed instead.

12. l. 155: 'during interglacial intervals of MIS 13 and 10'. This is just wrong: (i) MIS 13 is an interglacial, no need to write 'interglacial intervals of MIS 13', which implies that MIS 13 could also have glacial intervals; (ii) MIS 10 is a glacial.

We acknowledge that what was written was wrong and we changed the wording accordingly (l. 124).

13. l. 162-163: this is one peak recorded in only one sample (sample resolution 10 ky). Therefore I would be careful in interpreting this as a 'momentary return to seasonal sea ice cover'.

Thank you for your comment. We agree that this interpretation is only based on one data point and as such is rather too speculative, so we have taken it out.

14. The authors tend to over-interpret and draw to wide-scoping conclusions without offering convincing and strong arguments supporting their interpretations. The first sentence of this chapter speaks for the whole Ms: it is argued that 'variations of sea ice and open water biomarkers during the past 1.5 Ma demonstrate major changes in NH sea ice dynamics'. Their non-continuous analysis focus on three separated intervals and interpretations cannot be reliably extrapolated to the past 1.5 Ma.

We take this comment very seriously, and would like to split our response into two parts. Firstly, we would like to acknowledge that we do have a non-continuous record of sea ice change in the Bering Sea, representing three extended intervals of the Pleistocene that are representative of the pre-MPT 41-kyr G/IG cycles, the MPT itself, and the 100-kyr cycles of the post-MPT world. We understand that some of the wording in the original manuscript was misleading and extrapolation to the past 1.5 Ma is challenging. We thus made sure to change our wording to make it more clear for the reader that the record is discontinuous and additional research is needed to confirm sea ice changes over the past 1.5 Ma (see the abstract, l. 139-140, l. 292).

Secondly, despite the record being dis-continuous, we nonetheless identify significant changes in the Bering Sea sea ice regime across the sampled intervals. Specifically, we have produced records in order to explicitly test hypotheses of sea ice controlling glacial lengthening and therefore causing the MPT (SIS hypothesis, summarised on l. 23-35). For

this study, we therefore focussed on particular time intervals to test *predicted* change. In particular, our ‘middle’ MPT section (stretching over a 400 ka time period) shows a secular change in sea ice dynamics in models, which coincides with the first full 100 ka glacial. This interval of the MPT was studied at higher resolution, to comply with the main focus of this study. However, pre- and post-MPT sampling intervals are also of sufficient resolution to exemplify sea ice changes during these intervals. We are clear that our records, the first proxy data to explicitly test model-based hypotheses, are consistent and support conceptual/computer models for MPT development. Future studies may find changes in sea ice outside of our study intervals that disprove the SIS hypothesis, but we believe our study now provides enough evidence (proxy and model-based) to make a convincing case that BS sea ice plays a crucial role for glacial cycles, NPIW formation, and abyssal North Pacific carbon storage over the MPT.

- 15. Several issues are shortly introduced without any further discussion. An example of this is the short explanation on the decoupling between Brassicasterol, and HBI III and IP25 across the interim state (Fig. 3).**

We thank the reviewer and acknowledge this comment and provide a new discussion point on what the possible reasons are for a correlation/decoupling of brassicasterol and IP₂₅ (l. 156-172), which build on the biomarker outline given in the introduction (l. 43-63). One potential mechanism i.e. Pacific water inflow can in fact be tested through abundance of *N.seminae* in sediments of U1343, thus providing the opportunity to bring in further comparative data. Overall this has greatly helped to improve the flow of the discussion through comparison of multi-proxy data.

- 16. l. 219: it is incorrect to state that the low values of IP25 during MIS 22 ‘may reflect the relatively low sample resolution, especially across the termination.’ Actually eight samples are assigned to MIS 22, a higher amount than that of previous G.**

We thank the reviewer for this comment. We do agree that the glacial MIS 22 has a good sample coverage, and have therefore taken this statement out. We have replaced this with a more focused description of the potential reasons for lower IP₂₅ concentrations across MIS 22 and support our findings with additional data (U1343 biogenic opal accumulation; Fig. 4 and supplementary Fig. 2). (l. 186-192)

- 17. l. 220: I do not understand why the authors state ‘that the inverse relationship between IP25 and HBI III (Fig. 3) throughout the late Pleistocene indicates the renewed onset of pronounced seasonal sea ice cyclicity in the eastern Bering Sea’. Figure 3 presents results of the MPT, not temporally located in the late Pleistocene.**

We understand the confusion due to our wording. The late Pleistocene here means our pre-defined interval of ‘late Pleistocene sea ice dynamics’, which starts at 0.95 Ma, represented in Fig.3. However we have now changed the wording to avoid the ambiguity of ‘late Pleistocene’ to mid-to-late Pleistocene (l. 183).

- 18. l. 227: since there is a 300 ka-gap in the dataset between late MPT and MIS 13, it is misleading to state that 'MIS 12 is the first glacial throughout the entire record where perennial sea ice is encountered'. A few lines below is stated that 'perennial sea ice conditions are also recorded at the end of MIS 10 suggesting that this could be a common feature of past MPT climate cycles'. How can this be stated when the studied post-MPT interval is limited to a narrow time window of less than 200 ka (around the Mid-Brunhes Event) and an in-between 300 ka-gap exists?**

Thank you for the comment. We have changed our wording regarding MIS 12, as we acknowledge that this is simply the first glacial sampled with perennial sea ice conditions encountered throughout our three analysed intervals. The suggestion that perennial sea ice conditions could be a common feature of late Pleistocene glacial/interglacial cycles is supported by records for the last deglaciation from several marginal seas (l. 192-200).

- 19. l. 247: Terminations should be clearly identified in Figs. 2 and 4.**

This is a good point. However, we refrained from adding more vertical bars to the figures and instead marked the first termination where the deglacial sea ice peak occurs with an asterisk (see Fig. 2 and 4).

- 20. l. 248-250: This kind of statements (potential role of sea ice in increasing atmospheric CO₂ concentrations or change in sea ice dynamics as a consequence of Mid-Brunhes Event) is too vague.**

We thank the reviewer and acknowledge this comment. We agree that the link between ice free conditions and increased atmospheric CO₂ concentrations, although interesting to speculate on, is quite vague. Since we don't have sufficient data to support this link we decided to remove it, as we feel it is beyond the scope of the current manuscript. Instead we chose to focus more on the link of sea ice and potential atmospheric CO₂ drawdown during glacials (see answer to question no. 3, reviewer #2).

- 21. l. 259: how did the authors determine that the peak during late MIS 26 had a duration of 4 ka, when sample resolution is 5 ka for the MPT? (see l. 118).**

The sample resolution of 5 ka corresponds to the average sample resolution, however the sample spacing across the MPT interval is variable and MIS 26 actually has an average sample resolution of 3 ka with the sea ice peak at the end of MIS 26 itself having a sample resolution of 1 ka.

- 22. l. 262-280: it is true that the long-term cooling intensified around 1.1 Ma at Site 882 (roughly matching the interim state at Site 1343) and that the U1343 diatom record shows growing influence of the warm and salty Alaskan Stream into the interim state. However, SSTs at site 882 became warmer afterward, while the relative contribution of Neodenticula seminae (proxy for the Alaskan Stream) remained low. How this match the proposed 'renewed seasonality of sea ice' (l. 275)?**

The *N.seminae* record at U1343 shows decreased influence of warm and salty North Pacific waters after the interim state (<0.95 Ma) indicating that less inflow from the south has allowed for the sea ice to expand further across the Bering Sea, initiating renewed seasonality. The increase in ODP 882 SSTs is interpreted as a retraction of polar water masses, however U1343 is further north than ODP 882 so this may not have influenced the core site of U1343 (l. 173-182).

23. l. 295-300: The authors give the reader the impression that they are about to do more than just highlighting the potential of the Bering Sea to play a role in CO₂ drawdown during the MPT and afterward. Of course, the lack of a CO₂ record did not stop those working on Site 882 from speculating on the role of the Subarctic Ocean in carbon sequestration across the onset of NHG 2.7 Ma, but it is necessary to remember that we still do not know what happened to CO₂ across the MPT. Whatever happened to polar ocean CO₂ sequestration (see also l. 248-250), it is unlikely that have contributed to the origin of the MPT. The authors shortly address the link between CO₂ and ice free conditions, but this contributes neither to the main Ms message nor to solve the question on ice ages and CO₂ content in the atmosphere.

Please see answer to comment no. 20 and no. 3 (reviewer #2).

24. The main message of the Discussion subsection 3.3 seems to be the change in sea ice cover during the MPT was essential for the timing and behavior of late Pleistocene climate cycles. It does not add much new to the general discussion and does not offer a clear and focused 'take-home' message for the reader nor clearly present future perspectives in paleoclimatic reconstruction.

We thank the reviewer for these comments. The original intention was to break down the discussion into subsections to emphasize the different aspects of this manuscript, but we agree that 3.3 doesn't conform very well to this format. We revisited the importance of Bering Sea sea ice change for the SIS and rearranged the discussion to create a more coherent and unified conceptual model. In addition we removed the conclusion section and added a summarizing paragraph together with future perspectives and research opportunities (l. 287-297).

The discussion is now structured according to the following:

1. Introduction and overview of the three intervals of sea ice dynamics identified in the Bering Sea
2. The early-mid Pleistocene interval sea ice dynamics
3. The interim state sea ice dynamics
4. Implications of brassicasterol variability and the role of North Pacific inflow into the Bering Sea for sea ice dynamics
5. Late Pleistocene sea ice dynamics
6. Underlying mechanisms for changes in sea ice dynamics (comparison with global and regional climate records)

7. Implications for the sea ice switch model
8. Implications of sea ice change for GNPIW formation and glacial abyssal North Pacific carbon storage
9. Summary and future perspectives

25. Figure 2: there are several issues with this figure.

- I assume that light blue and light green(?) lines in 2c (IP25 and HBIII) represent the lower concentrations of IP25 and HBIII compared to those in 2B and 2A: this is not explained in the caption.

We modified Fig. 2 so that all three panels are consistent.

- The $\delta^{18}\text{O}_b$ -based stratigraphy in 2c is poorly resolved. The ups and downs hardly allow distinguishing between G and IG.

This is true. However, to date there are no further oxygen isotope records available from U1343. The $\delta^{18}\text{O}_b$ stratigraphy is, however, supported by alignment with bio- and magnetostratigraphic datums (see Asahi et al. 2016). Further, although we do make temporal assumptions about sea ice change on G/IG timescales across this interval, these are not pivotal for the interpretation and discussion of the data. Based on this we have changed the upper panel in Fig. 6 to represent the early MPT interval only.

- Why is LR04 Stack shown in 2C, but it is absent in 2B and 2A?

Please see answer to question 25.1

- 2A: the caption 'Marine isotope stage (MIS) 13 to 10 spanning the Mid Brunhes event' is misleading: the Mid-Brunhes event roughly occurred around 430 ka, matching the transition between MIS 12 and 11. It should be clearly identified in this figure.

We acknowledge this comment and have changed the caption accordingly.

- Why having two different y-axes for IP25 and HBIII in 2C? It is ok if concentrations are lower than in 2B and 2A, but having two axes for the same proxy with two different scales in the figure is unnecessary and confusing.

Please see answer to question 25.1

- 2B: why MPT is called here 'Mid-Pleistocene CLIMATE Transition' while is Mid-Pleistocene Transition throughout the MS?

Thank you for this comment. We made sure that the MPT is called the Mid-Pleistocene transition throughout the entire manuscript and figure captions.

Replies to reviewer's #3 comments:

1. The objects are missing in the introduction

We thank the reviewer for comments on the structure of the manuscript. This has greatly improved the flow of the text. Having made changes to the introduction in response to both reviewer #2 and #3 the structure of the introduction is now as follows:

1. 'Setting the scene' for the study, indicating why our study is important (l. 2-5)
2. The subarctic North Pacific sea ice regime (l. 6-15)
3. The MPT (l. 20-35)
4. Introduction of the material and the study (l. 36-42)
5. Introduction of the proxy (l. 43-63)
6. Summary of the most important outcomes of the study (l. 64-71)

Hopefully our objectives will be clearer now, after having made these changes.

2. Lines 58-86: in this paragraph, the core location setting (lines 62-66) and the introduction of biomarkers (lines 67-77) need to be restructured. The start of the biomarker introduction is kind of incoherent.

We thank the reviewer for this comment. We have added a break between the core location paragraph and the proxy description paragraph to make it easier for the reader to follow the introduction. Furthermore, we have re-written the biomarker introduction in line with our answer to comment no. 4 (reviewer #3) to give a more focused and pertinent overview of the use of biomarker proxies in this study (l. 43-63).

3. Lines 75-83: Belt et al. (2015, EPSL) and Smik et al. (2016, OG) demonstrated that the marginal ice zone is characterized with the elevated HBI III contents in the Barents and Norwegian Sea using surface-sediment calibration. However, HBI III is abundant in surface sediments from the North Iceland Shelf influenced by the drift ice (Cabedo-Sanz et al., 2016, QSR). Therefore, the authors should be careful with their interpretation due to the lack of surface-sediment calibration in the Bering Sea. This is very important for this manuscript.

We thank the reviewer for this in-depth comment. We agree with the reviewer that this is a very important set of points regarding this manuscript. The reviewer is correct regarding the Barents Sea data, and although the issue raised here is important, we might just point out that Cabedo-Sanz et al. (2016) reported the occurrence of HBI III in some surface sediments from the North Icelandic Shelf, but no relationship between abundance and sea ice was noted or discussed. However, we acknowledge that in any case there may have been drift ice present. The reviewer is of course completely correct that a surface sediment-based calibration has not yet been carried out for the Bering Sea. However, for the purpose of this study we note that the Barents Sea and the Bering Sea sea ice regime at the present day are very similar, with a pronounced seasonal advance and retreat of sea ice (l. 51).

- 4. Lines 97-98: to my knowledge, absent IP₂₅ along with low phytoplankton biomarkers should suggest perennial sea-ice condition. The wrong interpretation of biomarker data can mislead the discussion derived from the data set.**

We thank the reviewer very much for this comment. We also feel that this is an important point and we have re-considered the use of biomarkers for climatic interpretations in this study to address potential problems with using brassicasterol, as a result of the variety of its sources. In the original manuscript brassicasterol was used in support of HBI III for the interpretations of ice-free conditions, in line with previously published data (Müller et al. 2009, Smik and Belt 2017). Recent studies, however, demonstrate the potential difficulties with using brassicasterol due to its wide ranging sources and ambiguity of climatic interpretations (Belt et al. 2013, Navarro-Rodriguez et al. 2013). After further consideration and with regard to the positive correlation of brassicasterol and IP₂₅ between 1.53-1.36 Ma and 1.22-1.0 Ma (see supplementary Fig. 1), we cannot rule out a possible sea ice source for brassicasterol in the Bering Sea, at least during the earlier parts of the record. Having considered this, we elected to modify the use of brassicasterol and to retain it simply for interpretation of general phytoplankton production. This does not change the original interpretation of the sea ice regime in the eastern Bering Sea, as brassicasterol was originally used to support our interpretations based on HBI III. In fact it even adds a new and interesting point to the manuscript studying the correlation of brassicasterol and IP₂₅ and its implications (l.156-172). In line with these changes we have revised the biomarker introduction (l.43-63) to provide a more holistic and comprehensive description of the use of biomarkers for the climatic interpretations in this study. Additionally, we modified Fig. 2 and replaced the brassicasterol records (now found in supplementary Fig. 1) with the published biogenic opal record of Site U1343 (Kanematsu et al. 2013), which we also added to Fig. 4. Opal producing phytoplankton is believed to make up the majority of the primary productivity in the eastern Bering Sea and thus biogenic opal is interpreted to define first order changes in productivity at the eastern Bering slope (Kanematsu et al. 2013, Kim et al. 2014). Therefore, the biogenic opal record at Site U1343 aids in interpreting the sea ice regime, as intensive sea ice cover reduces the primary productivity due to light limitations, which can be seen in the weak negative correlation of IP₂₅ and biogenic opal (Fig. 5).

- 5. Lines 105-108: IP₂₅ and HBI III concentrations are generally during MIS 51-44 while brassicasterol concentrations are comparable to those during MIS 36-20 and MIS 13-10. The authors interpreted this as the degradation in this older section, but IP₂₅ should be stable than brassicasterol preserved in marine sediments.**

Thank you for your comment. In l. 83-85 we state that lowered HBI III and IP₂₅ concentrations across the oldest studied time interval could potentially indicate degradation, especially for HBI III, which has been shown to be more reactive compared to IP₂₅. However, in l. 85-88 we argue why we think degradation is unlikely. Although the reviewer suggests that IP₂₅ should be more stable than brassicasterol, we are not aware that has been demonstrated, definitively, in such sediments.

- 6. Lines 162-164: Why the HBI III concentration is still low during this momentary return of seasonal sea ice?**

The reason for this is not especially clear, but may be indicative of heavy sea ice cover, even if seasonal (see Belt et al., 2015 for similar observations in the Barents Sea). Additionally many data points across the three intervals with increased IP₂₅ concentration show low HBI III content, potentially indicating the different regimes these biomarkers are indicative for. Even though their regimes overlap under seasonal sea ice more extensive sea ice cover could lead to decrease HBI III concentrations in the sediment.

- 7. Lines 180-182 : the low/absent IP₂₅ in Arctic sediments is considered to reflect either ice free conditions or permanent ice cover as the authors mention in the introduction section. The predominantly ice-free conditions proposed by the authors here should be interpreted with the phytoplankton biomarker.**

We agree with this comment. Please see answer to question no. 4 (reviewer #3).

- 8. Lines 204-206: actually, the brassicasterol has a variety of sources, including marine and riverine sources, and even sea ice origin. Thus, the extended sea ice cover can not necessarily result in low brassicasterol concentrations. The brassicasterol concentrations in surface sediments from the Barents and Norwegian Sea are also abundant under the marginal ice zone as the same as HBI III (Belt et al., 2015, EPSL).**

We agree with this comment. Please see answer to question no. 4 (reviewer #3).

- 9. Lines 227-229: I only see low HBI III concentrations of MIS 12, but the concentrations of IP₂₅ and brassicasterol are variable. Cite references to support this assertion.**

Thank you very much for your comment, we realised that there was a typo in our original manuscript, missing the word 'late' prior to MIS 12 (l. 193).

- 10. Concerning the mechanisms of sea ice changes, the authors propose that an extensive sea ice cover causes the land glacier retreat via the temperature-precipitation mechanism. However, the response of sea ice to the climate change should also be taken into count.**

Thank you for your comment. It is true that we argue that extensive sea ice cover across glacial terminations has the ability to decrease continental glacier volume via the temperature-precipitation feedback. In the SIS hypothesis, essentially sea ice acts to precipitate a deglaciation, and then sea ice retreats rapidly at the end of a termination in response to global warming. However, we also argue that the changes in Bering Sea sea ice dynamics originate from global and regional climate cooling as seen in many records from the N.Pacific, N.Atlantic, and modelling studies (see Fig. 4, lines 207-245). Also see as per response to reviewer #2 (question no. 24) the discussion has now been thoroughly revised to make it more holistic.

11. Figure 4: the curves indicating IP₂₅ and HBI III should be labelled. Although Figure 4 has been cited in the discussion section several times, the interpretation is rather loose to the figure.

In Fig.4 we have separated the curves showing IP₂₅ and HBI III and labelled the axes either side with the according biomarker. Further we have referenced Fig. 4 more often in the discussion to provide a more coherent connection of the data and our interpretations.

Reviewers' comments:

Reviewer #3 (Remarks to the Author):

After the revision according to the reviewers' comments, this manuscript has been improved. However, there are still some major questions requiring carefully thought.

1. some inappropriate/wrong citations.

Line 73: Meheust et al. (2013, OG) reconstructed the modern sea-ice conditions in the Bering Sea via IP25 distribution and the major study area is ice free today. Thus, IP25 indication in Meheust et al. (2013, OG) is not appropriate in this manuscript as the IP25 record spanning the glacial interval.

Line 76: wrong citation as no HBI III mentioned in Navarro-Rodriguez et al. (2013, QSR).

Line 81: wrong citation as no HBI III mentioned in Mueller et al (2012, QSR).

2. the interpretation of IP25, HBI III and brassicasterol are not correct in some parts of the manuscript.

Lines 73-76: Within the Barents Sea, HBI III content is low in ice-free area and at summer ice edge, while brassicasterol content is variable in the study area

Lines 107-108: again, this interpretation is not correct. Absent IP25 with low HBI could also refer to permanent ice cover. In Belt et al. (2015, EPSL), the ice margin (both summer and winter) was focused in the Barents Sea and no further samples from the permanent ice cover. Thus, the authors cannot interpret absent IP25 with low HBI as ice-free condition by citing this article.

Lines 152-154: this interpretation is kind of conflict with lines 107-108. First understand the mechanism of how IP25 reflects the sea-ice conditions, then apply this proxy to reconstruct sea-ice condition.

Lines 185-188: the positive correlation of IP25 and brassicasterol not necessarily refer to the same sources of these two biomarkers. Brassicasterol can be produced by phytoplankton under the ice edge in the Arctic.

3. I was attracted by the revised introduction because "interactions of sea ice with nutrient supply, phytoplankton growth, circulation patterns, and air-sea gas exchange" and "implications for glacial North Pacific Intermediate Water formation, abyssal North Pacific carbon storage, and land-glacier retreat via the temperature-precipitation feedback" were mentioned. But I found loose link between these objects and the main text after reading this manuscript. The main objects and mechanism should be addressed and focused.

Reviewer #4 (Remarks to the Author):

Review by Ruediger Stein (Bremerhaven, 02 August 2017)

1. General comments

It is generally accepted that sea ice with its strong seasonal and interannual variability is a very critical component of the Arctic (and global climate) system that responds sensitively to changes in atmospheric circulation, incoming radiation, atmospheric and oceanic heat fluxes, as well as the hydrological cycle. Over the past three to four decades, coincident with global warming and

atmospheric CO₂ increase, Arctic sea ice has significantly decreased in its extent as well as in thickness. The processes causing these recent rapid sea ice and climate changes, however, are not fully understood and subject of intense scientific and societal debate. In this context, detailed proxy records of past sea ice variability may help to better understand processes controlling sea-ice and climate changes and, by this, to approve climate models for prediction of future climate change.

The present manuscript is dealing with this “hot topic” in (paleo)climate research. Furthermore, it presents the first detailed and direct proxy records of sea-ice changes across the Mid-Pleistocene Transition (MPT), a fundamental shift in frequency and amplitude in climate change from 41-ka to 100-ka glacial-interglacial cycles. The sea-ice data are discussed in relationship to glacial North Pacific Intermediate Water formation, abyssal North Pacific carbon storage and land-glacier retreat via the temperature-precipitation feedback. This paper is certainly of overall interest for the broader scientific community working on all aspects related to climate change. In general, I strongly support publication in Nature Communications. In my mind, however, still a major revision is needed before the manuscript can be accepted for publication (see more detailed comments below).

2. Sea-ice reconstruction using a biomarker approach (my personal view)

I would like to start with a few points dealing with basics and the interpretation of IP25, brassicasterol and HBI-III. As I did not get the first version of the manuscript I could unfortunately not do this at an earlier stage in the review process. Reviewer 3 of the first version, however, has also addressed several of my points (concerns), and I think not all the concerns/comments of Reviewer 3 have been considered in the revised manuscript.

First of all, I think, the IP25 approach developed by Belt et al. (2007) is certainly a milestone in research dealing with sea-ice reconstruction, and also the introduction of the PIP25 is an important further progress (Müller et al., 2011; Belt et al., 2015; Smik et al., 2016). But there are also difficulties to be considered when using this approach as also outlined in many papers (in a large number of these papers the co-authors of this paper are involved as well). As also outlined and discussed here, IP25 values of (around) zero are difficult to interpret (ice-free vs. permanent ice cover) without any additional parameter. Thus, IP25 and open-water phytoplankton biomarkers have been discussed together (Müller, Massé, Stein & Belt 2009) and even combined to the so-called PIP25 proxy (Müller et al., 2011). In the original work by Müller et al. (2011), brassicasterol and dinosterol have been used as phytoplankton biomarker. Here, there is a general agreement that the use of brassicasterol might be problematic as its source might be quite different, in my mind either marine or lacustrine/riverine (as shown by many studies), or – as proposed by Belt et al. (2013) – even sea ice. For a sea-ice origin, at least to my knowledge, the final proof is still missing (this would be the direct identification of brassicasterol in sea-ice algae species) as so far brassicasterol has only been identified in sea-ice samples together with IP25 but also other terrestrial higher-plant biomarkers. That means, brassicasterol and higher-plant biomarkers together (both with a terrestrial source!) might have been incorporated into sea ice in the marginal sea/marginal ice zone, contemporaneously with IP25 synthesized by the sea-ice algae, and then transported within the sea ice into/throughout the open ocean.

Furthermore, a stable ice-edge situation with ice melting and related nutrient and sediment release, is characterized by high concentrations of IP25, open-water phytoplankton biomarkers (brassicasterol, dinosterol, HBI-III) and also terrigenous biomarkers (β -sitosterol, long-chain n-alkanes, brassicasterol (!) as well as IRD). Under such conditions, there might/should be a positive correlation between IP25 and brassicasterol (or HBI-III) although the sources are different (sea ice vs. marine). That means, the interpretation of the different biomarker concentrations and their correlations (e.g., line 53-63, line 156 ff., Fig. 3, etc.) in terms of sea-ice concentrations/conditions is certainly difficult and should be done with caution. In this context, the introduction of the HBI-III as open-water/ice-edge biomarker (Belt et al., 2015; Smik et al., 2016)

is certainly an important further improvement of the biomarker approach for sea-ice reconstruction.

Concerning the interpretation of the IP25 and HBI-III records (Fig. 2), especially in the identification of ice-free conditions (pink bars in Fig. 2) and perennial sea ice conditions (blue bars in Fig. 2), I still have some major problems and cannot follow the argumentation of the authors (for me, it's looks partly even inconsistent!): Sometimes close to zero values of both IP25 and HBI-III are interpreted as ice free, sometimes as perennial sea ice (MIS 12 and 11) The argument for the different interpretation are the biogenic opal percentages that show values of about 15-20% and 8-10%, respectively. Both values indicate significant amount of opal indicative for primary production (especially if supported by opal accumulation rate data; see one further comment on this below), i.e., not perennial sea-ice coverage. Near 0.50 Ma (MIS 13) a minimum in IP25 correlates with a maximum in HBI-III (and biogenic opal of about 12%), also interpreted as "ice-free conditions" (pink bar) !? Near 0.88 Ma (MIS 22), 1.08 (MIS32/31) and 1.17 Ma (MIS 35), both IP25 and HBI-III are (close to) zero, biogenic values are 5% (!), 10% and 10%, respectively. All three intervals are interpreted "ice-free conditions" (pink bar)!? Two further examples are in MIS 47 and 49, i.e., both IP25 and HBI-III are zero, however, no opal data are available. Nevertheless, the biomarker data are interpreted again as "ice-free conditions" (pink bar)!? Furthermore, most of the intervals with "ice-free conditions" (pink bar) are just based on one (!) sample/data point each! Please have in mind, that many other data points show values close to zero for both biomarkers. For these data not discussed and highlighted in pink, however, sea-ice conditions might have been the same or at least quite similar. I myself would delete/exclude the pink bars! But more important for me: Please interpret the data more cautiously and consistently, and concentrate more on the general differences between glacial and interglacial intervals, differences between prior-MPT, MPT and post-MPT times, and the implications of Bering Sea sea-ice conditions for (regional to global) climate change.

3. Why no PIP25 data is included in this study?

Concerning the sea-ice reconstruction, I am wondering why – in addition to the single biomarker records - the authors have not produced "PIP25"-records. As outlined in Belt et al. (2015: "The potential for using a marker for semi-quantitative sea ice estimates using PIP25 (or related) index is quite attractive") and Smik et al. (2016), PIP25, especially after introducing the HBI-III as phytoplankton biomarker, gives important additional information about sea-ice conditions, as these authors showed very nicely in their studies of surface sediments and selected sediment cores from the Barents Sea. In addition, Belt et al. (2015) also demonstrated in the study of the sediment cores that calculated PIP25 indices using brassicasterol and HBI-III as phytoplankton markers yielded similar outcomes (!!) if core-specific c factors were used. As the authors by themselves have mentioned in their manuscript that sea ice conditions in the Bering Sea are quite similar to those in the Barents Sea, it surprises me even more that no PIP25 indices are presented and discussed here. Any argument for not using PIP25 in this study?

4. Biomarker data combined with biogenic opal and diatom data: Other records

For the interpretation of the biomarker data in terms of sea ice, open-water primary production and currents systems (here: Alaskan Stream inflow) it is very useful to consider other type of data, e.g., diatoms (*N. seminae*) and biogenic opal data, as also done in this paper. A similar approach with high-resolution IP25, biogenic opal, and diatoms records as well as direct comparisons (x-y plots) of IP25 with *N. seminae* and biogenic opal (see Fig. 5 of this paper) has been carried out for selected sediment cores from the Bering Sea, representing cold (Younger Dryas, HS1) and warm (Bölling/Alleröd, early Holocene) intervals of the last deglacial-Holocene transition (Méheust et al., 2015). Based on these data, sea-ice distribution maps for the western Bering Sea for cold and warm intervals have been produced. These data and maps should be considered in the discussion of this paper. A pdf file of this paper by Méheust et al. (2015) is submitted together with my review.

5. Results and discussion in general

Looking at the record of glacial/interglacial cycles and its time resolution, it is obvious that some glacial and interglacials are only represented by a very few data points. Thus, some of the general statements dealing with the shift in the timing of glacial IP25 maxima are quite vague. In stages 34, 30 and 28 there are no data points. On the other hand, in the glacial 26 and 24 the final IP25 maxima are based on one (!) data point. In my mind, the authors should more clearly mention in the discussion part that their results, i.e., the first MPT proxy record of sea ice – despite its low time resolution – gives strong evidence about the importance of Bering Sea sea ice for glacial cyclicity, NPIW formation etc. and is very useful to explicitly test model-based hypotheses for the MPT development (sea-ice switch hypothesis, etc.). This study is an important first step forward (“key pilot study”), although some of the statements remain still vague and speculative (this, however, I personally do not see as weak points as long as open points & questions are mentioned in the discussion part of the paper!). Thus, further studies have to follow now to finally approve the important findings of this study.

5. Other comments

Abstract, first line:

Add “role” -play an important role for both

Lines 46/47, 127 ff:

In the introduction, line 46/47, the absence of IP25 is related to ice-free conditions without mentioning that the other extreme (permanent sea ice) might be an explanation. Later in the text (line 127 ff) the other option is presented. Both options for the interpretation of IP25=0 should be already mentioned in the introduction.

Line 85: “biomarker degradation”

I do not see any real argument why the lower biomarker concentrations in the lower part of the record is related to biomarker degradation (“...potentially as a result of increased biomarker degradation”). For me it could be just an interval with lower concentration due to lower original production/flux.

Line 87:

These biomarkers were even found in late Miocene sediments (Stein et al., 2016).

Line 129:

The papers 25 (Xiao et al., 2015), 27 (Müller et al., 2009) and 43 (Stein and Fahl, 2012), are not dealing with HBI-III at all!

Line 188:

It might be useful to also plot the occurrence of sea-ice related diatoms from Site U1343 (see Kanematsu et al., Fig. 7; Teraishi et al., 2016, Fig. 7)? How do these records correlate with your biomarker records (IP25)?

Lines 189 and 201:

In this paper, biogenic opal concentrations are plotted in Figs. 2 and 4, and used as proxy for primary production. As outlined in Kanematsu et al. (2013), the opal percentage values are also strongly influenced by dilution due to terrigenous input. Thus, it might be better to plot the biogenic opal accumulation rates (see Kanematsu et al., Fig. 4) because they are not influenced by dilution but represent flux related to primary production. This would also be in line with the text (e.g., lines 189 and 201), where the term “opal accumulation rates” is used.

Check order of figures mentioned in the text:

Fig. 4 (line 153), Fig. 5 (line 190), Fig. 6 (line 145)

Check references:

Incomplete references No. 2, 8, and 55

Figures

Figs. 1 and 6

It might be useful to add locations of other sites from which data have been used, e.g. for reconstruction of sea-ice distribution for glacial (cold) and interglacial (warm) periods (Caissie et al., 2010; Max et al., 2012; Méheust et al., 2015).

Figs. 2 and 4:

Show sea-ice diatoms as well?

Fig. 4d:

The arrow (- < > +) might be misleading as IP25 values of (about) zero might be related to permanent sea ice cover!

Fig. 5b:

It might be useful (for comparison) to include IP25 and biogenic opal data from the Méheust et al. (2015) cores representing cold and warm periods?

Some additional important/useful references; at least (1) and (2) should be considered in this paper as well:

1. Méheust, M., Stein, R., Fahl, K., Max, L., Riethdorf, J.-R., 2015. High-resolution IP25-based reconstruction of sea-ice variability in the western North Pacific and Bering Sea during the past 18,000 years. *GeoMarine Letters*, DOI: 10.1007/s00367-015-0432-4.
2. Stein, R., Fahl, K., Schreck, M., Knorr, G., Niessen, F., Forwick, M., Gebhardt, C., Jensen, L., Kaminski, M., Kopf, A., Matthiessen, J., Jokat, W., and Lohmann, G., 2016. Evidence for ice-free summers in the late Miocene central Arctic Ocean. *Nature Communications* 7: 11148, doi:10.1038/ncomms11148.
3. Weckström, K., Massé, G., Collins, L.G., Hanhijärvi, S., Bouloubassi, I., Sicre, M.-A., Seidenkrantz, M.-S., Schmidt, S., Andersen, T.J., Andersen, M.L., Hill, B., Kuijpers, A., 2013. Evaluation of the sea ice proxy IP 25 against observational and diatom proxy data in the SW Labrador Sea. *Quat. Sci. Rev.* 79, 53-62.
4. Xiao, X., Stein, R., Fahl, K., 2015b. MIS 3 to MIS 1 temporal and LGM spatial variability in Arctic Ocean sea-ice cover: Reconstruction from biomarkers. *Paleoceanography* 30, doi: 10.1002/2015PA002814

Comment on: Review of "Changes in eastern Bering Sea sea ice dynamics across the Mid-Pleistocene climate transition" by Detlef et al, Nature Communications.

Reply to reviewer's #3 comments:

After the revision according to the reviewers' comments, this manuscript has been improved. However, there are still some major questions requiring carefully thought.

1. some inappropriate/wrong citations.

Thank you very much for your in depth comments and detailed revision of the manuscript regarding the references, we appreciate this very much.

Line 73: Meheust et al. (2013, OG) reconstructed the modern sea-ice conditions in the Bering Sea via IP25 distribution and the major study area is ice free today. Thus, IP25 indication in Meheust et al. (2013, OG) is not appropriate in this manuscript as the IP25 record spanning the glacial interval.

We appreciate your comment, however we decided to include the reference (Méheust et al. 2013) to show that IP₂₅ has been successfully applied in the study area of the eastern Bering Sea. We are aware that this is a study looking at surface sediments and thus recent sea ice conditions, however we think that it is important to show the difference between the seasonal sea ice and ice free environment in the Bering Sea as characterised by IP₂₅.

Line 76: wrong citation as no HBI III mentioned in Navarro-Rodriguez et al. (2013, QSR).

In line 54-55 we state 'Lower abundances of HBI III and variable abundances of other phytoplankton biomarkers are found in year-round ice free settings.' The reference Navarro-Rodriguez et al. (2013) in this case refers to 'other phytoplankton biomarkers' identified in

the Barents Sea. To make this clearer we have split the references and added them after the specific biomarker they refer to. (line 54-55)

Line 81: wrong citation as no HBI III mentioned in Mueller et al (2012, QSR).

Thank you very much for your comment, we appreciate this very much. We agree that this was the wrong citation, and have removed it accordingly. We also checked all other references, to make sure no additional mix-ups have occurred.

2. the interpretation of IP25, HBI III and brassicasterol are not correct in some parts of the manuscript.

Lines 73-76: Within the Barents Sea, HBI III content is low in ice-free area and at summer ice edge, while brassicasterol content is variable in the study area

The reviewer is correct that HBI III is elevated along the spring sea ice edge and low under ice free conditions, as stated in the manuscript, and brassicasterol is variable under ice free conditions with increased brassicasterol along the Norwegian coast in the Barents Sea. We have updated the manuscript accordingly to avoid any confusion (lines 54-55).

Lines 107-108: again, this interpretation is not correct. Absent IP25 with low HBI could also refer to permanent ice cover. In Belt et al. (2015, EPSL), the ice margin (both summer and winter) was focused in the Barents Sea and no further samples from the permanent ice cover. Thus, the authors cannot interpret absent IP25 with low HBI as ice-free condition by citing this article.

Please see response to reviewer #4 comment number 2

Lines 152-154: this interpretation is kind of conflict with lines 107-108. First understand the mechanism of how IP25 reflects the sea-ice conditions, then apply this proxy to reconstruct sea-ice condition.

Please see response to reviewer #4 comment number 2

Lines 185-188: the positive correlation of IP25 and brassicasterol not necessarily refer to the same sources of these two biomarkers. Brassicasterol can be produced by phytoplankton under the ice edge in the Arctic.

Thank you very much for this comment. We do argue in the manuscript that the correlation of brassicasterol and IP₂₅ can result from either the same source, or nutrient release during spring sea ice melting and brassicasterol formation during the spring sea ice edge bloom (line 185-186). Thus, we don't necessarily argue for the same source for both biomarkers, we merely mention this as one possible explanation, as we cannot exclude it based on recent publications, showing a potential sea ice source of brassicasterol (Belt et al. 2013).

3. I was attracted by the revised introduction because “interactions of sea ice with nutrient supply, phytoplankton growth, circulation patterns, and air-sea gas exchange” and “implications for glacial North Pacific Intermediate Water formation, abyssal North Pacific carbon storage, and land-glacier retreat via the temperature-precipitation feedback” were mentioned. But I found loose link between these objects and the main text after reading this manuscript. The main objects and mechanism should be addressed and focused.

Thank you very much for this comment. We are aware that our data cannot confirm all of the proposed mechanisms, however it shows strong indications that support a role of sea ice for North Pacific Intermediate Water formation, and for the first time a sea ice/land ice hysteresis is shown, with potential implications for the timing of late Pleistocene deglaciations. However, we have added sections on the uncertainties of our statements (see answer to reviewer #4 comment no. 5). Additionally, we have changed our wording to make sure, that the reader is aware of the limitations of our interpretations, especially with regard to abyssal North Pacific carbon storage, as this is highly speculative, and needs further work in order to confirm the role of NPIW for carbon sequestration and the impact of sea ice on CO₂ outgassing at the eastern Bering slope.

Comment on: Review of "Changes in eastern Bering Sea sea ice dynamics across the Mid-Pleistocene climate transition" by Detlef et al, Nature Communications.

Replies to reviewer's #4 comments:

1. General comments

'...This paper is certainly of overall interest for the broader scientific community working on all aspects related to climate change. In general, I strongly support publication in Nature Communications...'

We would like to thank you very much for your comment, on the overall value of our study, especially the acknowledgement of the importance to a wider audience, as we think this study could be important both for the 'proxy-community' and 'modelling-community' of paleoclimate research.

2. Sea-ice reconstruction using a biomarker approach (my personal view)

I would like to start with a few points dealing with basics and the interpretation of IP25, brassicasterol and HBI-III. As I did not get the first version of the manuscript I could unfortunately not do this at an earlier stage in the review process. Reviewer 3 of the first version, however, has also addressed several of my points (concerns), and I think not all the concerns/comments of Reviewer 3 have been considered in the revised manuscript.

First of all, I think, the IP25 approach developed by Belt et al. (2007) is certainly a milestone in research dealing with sea-ice reconstruction, and also the introduction of the PIP25 is an important further progress (Müller et al., 2011; Belt et al., 2015; Smik et al., 2016). But there are also difficulties to be considered when using this approach as also outlined in many papers (in a large number of these papers the co-authors of this paper are involved as well). As also outlined and discussed here, IP25 values of (around) zero are difficult to interpret (ice-free vs. permanent ice cover) without any additional parameter. Thus, IP25 and open-water phytoplankton biomarkers have been discussed together (Müller, Massé, Stein & Belt 2009) and even combined to the so-called PIP25 proxy (Müller et al., 2011). In the original work by Müller et al. (2011), brassicasterol and dinosterol have been used as phytoplankton biomarker. Here, there is a general agreement that the use of brassicasterol might be problematic as its source might be quite different, in my mind either marine or lacustrine/riverine (as shown by many studies), or – as proposed by Belt et al. (2013) – even sea ice. For a sea-ice origin, at least to my knowledge, the final proof is still missing (this would be the direct identification of brassicasterol in sea-ice algae species) as so far brassicasterol

has only been identified in sea-ice samples together with IP25 but also other terrestrial higher-plant biomarkers. That means, brassicasterol and higher-plant biomarkers together (both with a terrestrial source!) might have been incorporated into sea ice in the marginal sea/marginal ice zone, contemporaneously with IP25 synthesized by the sea-ice algae, and then transported within the sea ice into/throughout the open ocean.

Furthermore, a stable ice-edge situation with ice melting and related nutrient and sediment release, is characterized by high concentrations of IP25, open-water phytoplankton biomarkers (brassicasterol, dinosterol, HBI-III) and also terrigenous biomarkers (β -sitosterol, long-chain n-alkanes, brassicasterol (!) as well as IRD). Under such conditions, there might/should be a positive correlation between IP25 and brassicasterol (or HBI-III) although the sources are different (sea ice vs. marine). That means, the interpretation of the different biomarker concentrations and their correlations (e.g., line 53-63, line 156 ff., Fig. 3, etc.) in terms of sea-ice concentrations/conditions is certainly difficult and should be done with caution. In this context, the introduction of the HBI-III as open-water/ice-edge biomarker (Belt et al., 2015; Smik et al., 2016) is certainly an important further improvement of the biomarker approach for sea-ice reconstruction.

Concerning the interpretation of the IP25 and HBI-III records (Fig. 2), especially in the identification of ice-free conditions (pink bars in Fig. 2) and perennial sea ice conditions (blue bars in Fig. 2), I still have some major problems and cannot follow the argumentation of the authors (for me, it's looks partly even inconsistent!): Sometimes close to zero values of both IP25 and HBI-III are interpreted as ice free, sometimes as perennial sea ice (MIS 12 and 11) The argument for the different interpretation are the biogenic opal percentages that show values of about 15-20% and 8-10%, respectively. Both values indicate significant amount of opal indicative for primary production (especially if supported by opal accumulation rate data; see one further comment on this below), i.e., not perennial sea-ice coverage. Near 0.50 Ma (MIS 13) a minimum in IP25 correlates with a maximum in HBI-III (and biogenic opal of about 12%), also interpreted as "ice-free conditions" (pink bar)!? Near 0.88 Ma (MIS 22), 1.08 (MIS32/31) and 1.17 Ma (MIS 35), both IP25 and HBI-III are (close to) zero, biogenic values are 5% (!), 10% and 10%, respectively. All three intervals are interpreted "ice-free conditions" (pink bar)!? Two further examples are in MIS 47 and 49, i.e., both IP25 and HBI-III are zero, however, no opal data are available. Nevertheless, the biomarker data are interpreted again as "ice-free conditions" (pink bar)!? Furthermore, most of the intervals with "ice-free conditions" (pink bar) are just based on one (!) sample/data point each! Please have in mind, that many other data points show values close to zero for both biomarkers. For these data not discussed and highlighted in pink, however, sea-ice conditions might have been the same or at least quite similar. I myself would delete/exclude the pink bars! But more important for me: Please interpret the data more cautiously and consistently, and concentrate more on the general differences between glacial and interglacial intervals, differences between prior-MPT, MPT and post-MPT times, and the implications of Bering Sea sea-ice conditions for (regional to global) climate change.

Thank you very much for this in depth discussion and review of the biomarker approach for sea ice reconstruction used in our manuscript. We appreciate your comments, and have considered them carefully.

We agree with the reviewer that the biomarker approach for sea ice reconstructions using IP₂₅, has advanced greatly in the past few years by combining IP₂₅ with phytoplankton biomarkers. However, as the reviewer points out uncertainties exist with regard to the sources of certain biomarkers, for example brassicasterol. Brassicasterol, has been shown to originate from marine (Goad et al. 1982), lacustrine (Volkman 1986), and potentially even sea ice sources (Belt et al. 2013), complicating its use as an open water phytoplankton biomarker. We thus decided to exclude brassicasterol from interpretations of the sea ice regime in the eastern Bering Sea, and use it as an indicator of general phytoplankton production instead. Thus, a positive correlation between IP₂₅ and brassicasterol (as observed during the oldest interval, and the earlier part of the MPT interval) could point towards

either a sea ice source of brassicasterol, or the importance of nutrient release during spring sea ice melting for phytoplankton productivity in the region (line 180-197).

In order to interpret the sea ice regime we use a combinatory approach of IP₂₅ (indication of seasonal sea ice), HBI III (most prominent in the ice marginal zone), and biogenic opal (first order changes in export productivity), measured in IODP Site U1343 in the eastern Bering Sea. The biomarker IP₂₅ is an indicator of seasonal sea ice, whereas low or zero IP₂₅ can originate from both ice free and perennial sea ice conditions. Recently, IP₂₅ has been found in regions of permanent/near permanent sea ice cover (Xiao et al. 2015), indicating that small amounts of IP₂₅ can be present in sediments underlying perennial sea ice. Thus in the original biomarker approach, zero IP₂₅ values were interpreted as ice free, whereas very low amount of IP₂₅ (partly even below the limit of quantification), were interpreted as a more extended ice cover. As HBI III is increased under ice marginal zone conditions, both, ice free and perennial sea ice conditions, results in lowered HBI III accumulation, making it difficult to distinguish between the possible sea ice scenarios based on IP₂₅ and HBI III alone. Thus, we also used biogenic opal wt% across the MPT and the youngest interval to aid with the interpretation of the sea ice regime.

However, we have considered the reviewers comments carefully and have developed a more systematic approach to interpreting the sea ice regime, following the example of Méheust et al. 2016, who studied sea ice in the western Bering Sea over the last 20 ka. Additionally, as suggested by the reviewer we are using the mass accumulation rate of biogenic opal (MAR_{opal}) rather than biogenic opal (wt%) in Site U1343, as the MAR_{opal} is more representative of export productivity and excludes any influence of dilution by terrigenous sediment input. The MAR_{opal} record (Kim et al. 2014) also goes back further in time, allowing us to compare all three reconstructed intervals to fluxes of export productivity. Méheust et al. 2016 use cross-plots of IP₂₅/biogenic opal and IP₂₅ together with certain diatom species indicative of warm and cold surface waters, respectively, to characterise the sea ice regime in the western Bering Sea. The IP₂₅ concentrations measured in the western Bering Sea across the past 20 ka are higher compared to the eastern Bering Sea Pleistocene intervals. This could be related to either the sea ice regime/seasonal duration of sea ice over the study location, or it could be indicative of an increase in sea ice extent/duration during the late Pleistocene. Building on the approach of Méheust et al. 2016, we have developed a multi-proxy analysis that combines IP₂₅, HBI III, and MAR_{opal} in Site U1343 to classify the sea ice state (see Methods, line 371- 401). We use threshold values of IP₂₅, HBI III, and MAR_{opal} to distinguish between four different sea ice regimes.

Table 1. Boundaries for identification of sea ice states

Sea ice state	IP ₂₅ (ng g ⁻¹ sed)	HBI III (ng g ⁻¹ sed)	MAR _{opal} (g cm ⁻² ka ⁻¹)
Ice free	< 0.5	< 0.7	> 4
Extended sea ice	< 0.5	< 0.7	< 4
Seasonal sea ice (within the MIZ)	> 0.5	> 0.7	variable
Seasonal sea ice (outside the MIZ)	> 0.5	< 0.7	variable

The threshold values for IP₂₅ and HBI III are based on 10% of the total range of the respective biomarker and summarised in Supplementary Fig. 1 and Table 1), while the corresponding value for MAR_{opal} is based on the past 20 ka. Both, Méheust et al. 2016 and Caissie et al. 2010 (diatom assemblage sea ice reconstruction from the Umnak Plateau), suggest a change from perennial to ice free/ seasonal sea ice conditions in the Bering Sea at around 15 ka BP. The MAR_{opal} in Site U1343 at 15 ka is at 4 g cm⁻² ka⁻¹ and we use this as our threshold to distinguish between perennial sea ice and

more seasonal sea ice/ice free conditions (Fig. 5, Supplementary Fig. 1). Seasonal sea ice conditions in the eastern Bering Sea show variable MAR_{opal} . Three data points do not fit with this classification (Table 2), as they have HBI III values outside of our threshold for extended/ice-free scenarios. The reasons for this are unclear, but potentially indicates predominantly ice free/perennial sea ice conditions with occasional marginal ice zone sedimentation. The MAR_{opal} of these data points is used to distinguish between ice free and extended ice cover scenarios (Table 1).

This systematic approach to interpret the sea ice regime, largely fits with our original interpretation apart from a few data points during the oldest interval and the earliest MPT interval, indicative of extended ice cover, rather than ice free conditions. However, dating uncertainties prior to 1.2 Ma, hinder the interpretation of the glacial/interglacial development of the sea ice regime in the eastern Bering Sea. Additionally, early interglacial MIS 25 is also characterised by an extended ice cover, which supports our observations of late glacial/termination sea ice peaks in the eastern Bering Sea during MPT and post MPT climate cycles. Following this approach, we have removed the pink and blue bars, indicative of ice free and extended sea ice cover, respectively, from the figures. As a supplementary figure we have included Fig. 4 from the main manuscript, with IP_{25} data points coloured according to the sea ice scenario (Supplementary Fig. 1), and including the threshold values for all three proxies as horizontal lines, to visually demonstrate the validity of this approach (Fig. 4, Supplementary Fig. 1).

3. Why no PIP25 data is included in this study?

Concerning the sea-ice reconstruction, I am wondering why – in addition to the single biomarker records - the authors have not produced “PIP25”-records. As outlined in Belt et al. (2015: “The potential for using a marker for semi-quantitative sea ice estimates using PIP25 (or related) index is quite attractive”) and Smik et al. (2016), PIP25, especially after introducing the HBI-III as phytoplankton biomarker, gives important additional information about sea-ice conditions, as these authors showed very nicely in their studies of surface sediments and selected sediment cores from the Barents Sea. In addition, Belt et al. (2015) also demonstrated in the study of the sediment cores that calculated PIP25 indices using brassicasterol and HBI-III as phytoplankton markers yielded similar outcomes (!) if core-specific c factors were used. As the authors by themselves have mentioned in their manuscript that sea ice conditions in the Bering Sea are quite similar to those in the Barents Sea, it surprises me even more that no PIP25 indices are presented and discussed here. Any argument for not using PIP25 in this study?

Thank you very much for your comment, we appreciate that PIP_{25} has experienced recurrent attention in the past years, and that Smik et al. (2016), demonstrate the improvement of the PIP_{25} index by using HBI III compared to other biomarkers, such as brassicasterol or dinosterol. However, difficulties exist with calculating the PIP_{25} index, as it includes a balance term (c-factor) to account for differences in the abundance of IP_{25} and phytoplankton biomarkers. The c-factor can be calculated from the downcore data, or from a regional surface sediment calibration. However, the latter does not yet exist for the Bering Sea. Calculating the c-factor based on mean sedimentary abundances of IP_{25} and the phytoplankton biomarker, however, potentially introduces bias due to the variability of the c-term based on the core section studied (Belt and Müller, 2013). Especially, with regard to the three core sections studied, the c-factor varies greatly for the individual intervals (MIS 13-MIS 10: 1.5, MIS 37-20: 0.94, MIS 51-44: 4.2), potentially also as a result of the difference in resolution and data points per glacial/interglacial intervals. Using a fixed c-factor for all three studied intervals, on the other hand, also introduces potential bias, based on the fact that we have a non-continuous record and variable resolution across the studied intervals.

Additionally, for intervals of zero IP_{25} and low HBI III that coincide with low biogenic opal MAR, and thus indicate an extended sea ice cover, PIP_{25} values are zero, indicating ice free conditions.

Therefore, we have refrained from including the PIP_{25} record in our manuscript, as it complicates the sea ice regime interpretation, and potentially introduces bias with regard to a semi-quantitative sea ice interpretation. However, in the future, such an approach could be used once a calibration (e.g. from surface sediments) has been undertaken.

4. Biomarker data combined with biogenic opal and diatom data: Other records

For the interpretation of the biomarker data in terms of sea ice, open-water primary production and currents systems (here: Alaskan Stream inflow) it is very useful to consider other type of data, e.g., diatoms (*N. seminae*) and biogenic opal data, as also done in this paper. A similar approach with high-resolution IP_{25} , biogenic opal, and diatoms records as well as direct comparisons (x-y plots) of IP_{25} with *N. seminae* and biogenic opal (see Fig. 5 of this paper) has been carried out for selected sediment cores from the Bering Sea, representing cold (Younger Dryas, HS1) and warm (Bölling/Alleröd, early Holocene) intervals of the last deglacial-Holocene transition (Méheust et al., 2015). Based on these data, sea-ice distribution maps for the western Bering Sea for cold and warm intervals have been produced. These data and maps should be considered in the discussion of this paper. A pdf file of this paper by Méheust et al. (2015) is submitted together with my review.

We particularly appreciate this comment, and have included the sea ice interpretation across the past 20 ka from Méheust et al. (2016) in our manuscript. As outlined above we use the LGM-Holocene data from Méheust et al. (2016), in addition to other sea ice studies across this transition, to define the boundary for MAR_{opal} under perennial and seasonal/ice free conditions in Site U1343. Additionally, we used the findings from the paper to support our interpretations regarding an extensive sea ice cover during the late glacial/termination (line 174-177) and have included the interpretation in our sea ice map of the late Pleistocene (Fig. 5).

5. Results and discussion in general

Looking at the record of glacial/interglacial cycles and its time resolution, it is obvious that some glacials and interglacials are only represented by a very few data points. Thus, some of the general statements dealing with the shift in the timing of glacial IP_{25} maxima are quite vague. In stages 34, 30 and 28 there are no data points. On the other hand, in the glacials 26 and 24 the final IP_{25} maxima are based on one (!) data point. In my mind, the authors should more clearly mention in the discussion part that their results, i.e., the first MPT proxy record of sea ice – despite its low time resolution – gives strong evidence about the importance of Bering Sea sea ice for glacial cyclicity, NPIW formation etc. and is very useful to explicitly test model-based hypotheses for the MPT development (sea-ice switch hypothesis, etc.). This study is an important first step forward (“key pilot study”), although some of the statements remain still vague and speculative (this, however, I personally do not see as weak point as long as open points & questions are mentioned in the discussion part of the paper!). Thus, further studies have to follow now to finally approve the important findings of this study.

Thank you very much for this comment. As this study is the first of its kind, looking at sea ice dynamics across the MPT, we are aware that some of the statements made need further

confirmation, and we appreciate your comment very much. To make sure the future readers are aware of the limitations of our data interpretation we have added sections with recommendations for future work to the discussion part of the manuscript (line 172-174, 234-236, and line 292-297, line 302-309).

5. Other comments

Abstract, first line:

Add “role” -play an important role for both

We have added ‘role’ to the first sentence of the abstract, thank you very much for spotting this typo.

Lines 46/47, 127 ff:

In the introduction, line 46/47, the absence of IP25 is related to ice-free conditions without mentioning that the other extreme (permanent sea ice) might be an explanation. Later in the text (line 127 ff) the other option is presented. Both options for the interpretation of IP25=0 should be already mentioned in the introduction.

We have modified our biomarker introduction to fit with the more systematic approach of sea ice regime interpretation we have adopted according to the reviewers comment no. 2 (line 44-64).

Line 85: “biomarker degradation”

I do not see any real argument why the lower biomarker concentrations in the lower part of the record is related to biomarker degradation (“...potentially as a result of increased biomarker degradation). For me it could be just an interval with lower concentration due to lower original production/flux.

We argued in the manuscript, that it could be a result of degradation, however, we are of the same opinion as the reviewer, that degradation is unlikely due to fairly constant TOC content in U1343. However, along with the reviewer, we decided that this is not of primary relevance to the manuscript and have deleted this section.

Line 87:

These biomarkers were even found in late Miocene sediments (Stein et al., 2016).

As mentioned above, we have deleted this section from the manuscript, however we have added a sentence to the introduction regarding the time interval, where IP₂₅ has been identified (line 45-47).

Line 129:

The papers 25 (Xiao et al., 2015), 27 (Müller et al., 2009) and 43 (Stein and Fahl, 2012), are not dealing with HBI-III at all!

The reviewer is correct, we have deleted these references, and have checked all of our references carefully, to make sure that all citations are now correct/accurate.

Line 188:

It might be useful to also plot the occurrence of sea-ice related diatoms from Site U1343 (see Kanematsu et al., Fig. 7; Teraishi et al., 2016, Fig. 7)? How do these records correlate with your biomarker records (IP25)?

Thank you very much for this comment. The sea ice diatom record and the IP₂₅ record from Site U1343 are positively correlated (pearson r correlation coefficient: $r_{xy} = 0.324$ with 95% Student's t confidence intervals [0.095; 0.521], $n = 142$), indicating that increased seasonal sea ice cover is marked by both increased sea ice diatom abundance and IP₂₅. However, as the sea ice diatom record is of much lower resolution (~15 ka on average), compared to the IP₂₅ record (~5 ka on average) we decided not to add the correlation to the manuscript, as we would potentially bias the sea ice diatom data, when re-sampling it at the same resolution as the IP₂₅ record. Instead, we added the sea ice diatom record (Teraishi et al. 2016) to Supplementary Fig. 1, for comparison. We also make a reference in the manuscript, to make the reader aware of this record (line 33-37).

Lines 189 and 201:

In this paper, biogenic opal concentrations are plotted in Figs. 2 and 4, and used as proxy for primary production. As outlined in Kanematsu et al. (2013), the opal percentage values are also strongly influenced by dilution due to terrigenous input. Thus, it might be better to plot the biogenic opal accumulation rates (see Kanematsu et al., Fig. 4) because they are not influenced by dilution but represent flux related to primary production. This would also be in line with the text (e.g., lines 189 and 201), where the term “opal accumulation rates” is used.

Thank you very much for your comment, this is a good suggestion. However, instead of using the Kanematsu et al. 2013 biogenic opal data, we now use the Kim et al. 2014 MAR_{opal} in Site U1343, especially as it extends further back in time. Thus, we are able to compare all three time intervals to MAR_{opal} and avoid potential bias due to dilution of terrigenous sediments.

Check order of figures mentioned in the text:

Fig. 4 (line 153), Fig. 5 (line 190), Fig. 6 (line 145)

We have changed the order of the figures, to conform to the order at which they occur in the manuscript.

Check references:

Incomplete references No. 2, 8, and 55

Thank you very much for this comment. We have changed the references mentioned, and checked all other references in EndNote, to make sure they are imported correctly into the manuscript.

Figures

Figs. 1 and 6

It might be useful to add locations of other sites from which data have been used, e.g. for reconstruction of sea-ice distribution for glacial (cold) and interglacial (warm) periods (Caissie et al., 2010; Max et al., 2012; Méheust et al., 2015).

Good idea. We have added the core locations of additional sea ice studies in the Bering Sea to Fig. 1 and the late Pleistocene maps of Fig. 5.

Figs. 2 and 4:

Show sea-ice diatoms as well?

As mentioned above, we refrain from showing the sea ice diatom record in the main manuscript figures, as it is of considerably lower resolution than the IP₂₅ record and thus does not add to the story of the manuscript. However, we added the record to Supplementary Fig. 1 for comparison and to show the generally good agreement.

Fig. 4d:

The arrow (- < > +) might be misleading as IP₂₅ values of (about) zero might be related to permanent sea ice cover!

We have adjusted the arrows in Fig. 4 to point out the different sea ice states, as demonstrated by the respective biomarker concentrations.

Fig. 5b:

It might be useful (for comparison) to include IP₂₅ and biogenic opal data from the Méheust et al. (2015) cores representing cold and warm periods?

Thank you for your comments, however we decided not to include the data, as the concentrations of IP₂₅ are very different in the Méheust et al. 2016 study. Additionally we now use MAR_{opal}, rather than biogenic opal (wt%), as used in the Méheust et al. 2016 study.

Some additional important/useful references; at least (1) and (2) should be considered in this paper as well:

1. Méheust, M., Stein, R., Fahl, K., Max, L., Riethdorf, J.-R., 2015. High-resolution IP₂₅-based reconstruction of sea-ice variability in the western North Pacific and Bering Sea during the past 18,000 years. *GeoMarine Letters*, DOI: 10.1007/s00367-015-0432-4.

2. Stein, R., K. Fahl, Schreck, M., Knorr, G., Niessen, F., Forwick, M., Gebhardt, C., Jensen, L., Kaminski, M., Kopf, A., Matthiessen, J., Jokat, W., and Lohmann, G., 2016. Evidence for ice-free summers in the late Miocene central Arctic Ocean. *Nature Communications* 7:11148, doi:10.1038/ncomms11148.

3. Weckström, K., Massé, G., Collins, L.G., Hanhijärvi, S., Bouloubassi, I., Sicre, M.-A., Seidenkrantz, M.-S., Schmidt, S., Andersen, T.J., Andersen, M.L., Hill, B., Kuijpers, A., 2013. Evaluation of the sea

ice proxy IP 25 against observational and diatom proxy data in the SW Labrador Sea. *Quat Sci. Rev.* 79, 53-62.

4. Xiao, X., Stein, R., Fahl, K., 2015b. MIS 3 to MIS 1 temporal and LGM spatial variability in Arctic Ocean sea-ice cover: Reconstruction from biomarkers. *Paleoceanography* 30, doi: 10.1002/2015PA002814

Thank you very much for these suggestions, we have included the first two studies in the manuscript (see above answers).

REVIEWERS' COMMENTS:

Reviewer #4 (Remarks to the Author):

Review by Ruediger Stein (Bremerhaven, 21 October 2017)

The authors have done a very careful revision of the manuscript. My main comments concerning text and figures have mostly been considered. For me it is most important that the authors interpret the data more cautiously and consistently, and concentrate more on the general differences between glacial and interglacial intervals, differences between prior-MPT, MPT and post-MPT times, and the implications of Bering Sea sea-ice conditions for (regional to global) climate change. I like very much that the authors have included the biogenic opal accumulation rate data (cf., approach by Méheust et al. 2015) for interpreting the biomarker data in terms of sea ice distribution and productivity and for distinguishing different sea ice scenarios/situations. Some of the statements remain still vague and speculative which, however, must not be a weak point as long as open points & questions are mentioned in the discussion part of the paper (see comment in my review of the earlier version). This has been done by the authors themselves in their revised version, i.e., they concluded that future (multi-disciplinary) studies have to follow to unravel the importance of Bering Sea sea ice for glacial cyclicity, NPIW formation, deep ocean carbon storage, etc.. Thus, from my point of view the manuscript is more or less "ready to go"!

There are only two points where I still have some (minor) criticism:

We all agree that it is difficult to identify the origin of the biomarker brassicasterol, and in several papers marine, terrigenous/lacustrine and sea-ice sources have been discussed. Whereas this biomarker has been determined directly in marine and lacustrine organisms (algae), it has not been identified in sea-ice algae so far. It has only been found in sea-ice samples that contain biomarkers of different origins (i.e., marine, terrigenous/lacustrine and sea-ice sources) as well as siliciclastic material (i.e., clay minerals, IRD, etc.). This means at least for me, the statement dealing with sea ice as source for brassicasterol with reference to Belt and Müller (2013) (e.g., line 203/204) is misleading. In this paper as well as in Belt et al. (2013) there is no real prove at all for a sea-ice origin. I propose to express this point more precisely.

In the manuscript it has been mentioned several times that there are high (5 ka) and low (10 ka, 8 ka) time resolution intervals. I personally would not use this strict classification scheme. How significant are differences between 10 and 8 or 8 and 5 ka? There are intervals with higher and lower time resolution, but there are also intervals represented by only a very few samples. These facts should be considered in the interpretation of the records.

Replies to reviewer's #4 comments:

'The authors have done a very careful revision of the manuscript. My main comments concerning text and figures have mostly been considered...'

We would like to thank you for acknowledging the work that has gone in to improving the quality of our manuscript following your previous review.

'We all agree that it is difficult to identify the origin of the biomarker brassicasterol, and in several papers marine, terrigenous/lacustrine and sea-ice sources have been discussed. Whereas this biomarker has been determined directly in marine and lacustrine organisms (algae), it has not been identified in sea-ice algae so far. It has only been found in sea-ice samples that contain biomarkers of different origins (i.e., marine, terrigenous/lacustrine and sea-ice sources) as well as siliciclastic material (i.e., clay minerals, IRD, etc.). This means at least for me, the statement dealing with sea ice as source for brassicasterol with reference to Belt and Müller (2013) (e.g., line 203/204) is misleading. In this paper as well as in Belt et al. (2013) there is no real proof at all for a sea-ice origin. I propose to express this point more precisely.'

Thank you very much for your comment. We acknowledge that, in some cases, brassicasterol may be found in sea ice as a result of entrainment of non-sea ice algal material (e.g. riverine input, particulate terrestrial material, IRD), which is why on line 75 and 184/185– we carefully state that brassicasterol may *potentially* be sourced from sea ice algae. To comply with a more careful interpretation of brassicasterol sources, we have added the uncertainty of non-biogenic entrainment of brassicasterol in sea ice to our interpretation in line 191-192. We agree, however, that the cited reference (Belt and Mueller 2013) was misleading and we have replaced the reference with Belt et al 2013 Organic geochemistry. It is feasible that the brassicasterol in that study had a non-biogenic origin (e.g. IRD) but that does not explain the bloom seen for this biomarker alongside that seen for IP₂₅, fatty acids and chlorophyll a. Coupled with our more careful wording regarding brassicasterol sources, we think that even though our study does not provide a definitive proof of a sea ice source for brassicasterol our interpretation still stands.

'In the manuscript it has been mentioned several times that there are high (5 ka) and low (10 ka, 8 ka) time resolution intervals. I personally would not use this strict classification scheme. How

significant are differences between 10 and 8 or 8 and 5 ka? There are intervals with higher and lower time resolution, but there are also intervals represented by only a very few samples. These facts should be considered in the interpretation of the records.'

We appreciate your comment. We agree that the difference in average resolution between the measured time intervals is not significant and have removed statements referring to the average resolution from the results section (line 96, 105, and 136). The statement regarding 10 ka resolution, refers to the benthic oxygen isotope curve and the age model uncertainties for Site U1343 (line 90). The variable sample resolution is considered in our discussion, with respect to the changes in the timing of the sea ice maxima across the Mid-Pleistocene transition (line 170, 235).